# CFAP53 regulates mammalian cilia-type motility patterns through differential localization and recruitment of axonemal dynein components

**Takahiro Ide** [1☯]*, **Wang Kyaw Twan** [1,2☯], **Hao Lu** [3☯], **Yayoi Ikawa** [1], **Lin-Xenia Lim** [3¤b], **Nicole Henninger** [2], **Hiromi Nishimura** [1], **Katsuyoshi Takaoka** [2], **Vijay Narasimhan** [3¤a], **Xiumin Yan** [4], **Hidetaka Shiratori** [2], **Sudipto Roy** [3,5,6]*, **Hiroshi Hamada** [1,2]*

1 Laboratory for Organismal Patterning, RIKEN Center for Biosystems Dynamics Research, Kobe, Hyogo, Japan, 2 Developmental Genetics Group, Graduate School of Frontier Biosciences, Osaka University, Suita, Osaka, Japan, 3 Institute of Molecular and Cell Biology, Proteos, Singapore, 4 Institute of Biochemistry and Cell Biology, Shanghai Institutes for Biological Sciences, Chinese Academy of Sciences, Shanghai, China, 5 Department of Biological Sciences, National University of Singapore, Singapore, 6 Department of Pediatrics, Yong Loo Ling School of Medicine, National University of Singapore, Singapore

☯ These authors contributed equally to this work.
¤a Current address: Zebrafish Centre for Advanced Drug Discovery and Keenan Research Centre for Biomedical Science, Li Ka Shing Knowledge Institute, St. Michael's Hospital, University of Toronto, Toronto, Ontario, Canada
¤b Current address: Graduate School of Frontier Biosciences, Osaka University, Suita, Osaka, Japan
* takahiro.ide@riken.jp (TI); sudipto@imcb.a-star.edu.sg (SR); hiroshi.hamada@riken.jp (HH)

**Data Availability Statement:** All relevant data are within the manuscript and its Supporting Information files.

## Abstract

Motile cilia can beat with distinct patterns, but how motility variations are regulated remain obscure. Here, we have studied the role of the coiled-coil protein CFAP53 in the motility of different cilia-types in the mouse. While node (9+0) cilia of *Cfap53* mutants were immotile, tracheal and ependymal (9+2) cilia retained motility, albeit with an altered beat pattern. In node cilia, CFAP53 mainly localized at the base (centriolar satellites), whereas it was also present along the entire axoneme in tracheal cilia. CFAP53 associated tightly with microtubules and interacted with axonemal dyneins and TTC25, a dynein docking complex component. TTC25 and outer dynein arms (ODAs) were lost from node cilia, but were largely maintained in tracheal cilia of *Cfap53*⁻/⁻ mice. Thus, CFAP53 at the base of node cilia facilitates axonemal transport of TTC25 and dyneins, while axonemal CFAP53 in 9+2 cilia stabilizes dynein binding to microtubules. Our study establishes how differential localization and function of CFAP53 contributes to the unique motion patterns of two important mammalian cilia-types.

## Author summary

Motile cilia in various kinds of tissues and cell-types drive fluid flow over epithelia or facilitate cellular locomotion. There are two types of motile cilia. Motile cilia with a 9+2 configuration of microtubules are found on tracheal epithelial cells and brain ependymal

**Funding:** This work was supported by funds from the Agency for Science, Technology and Research (A*STAR) of Singapore (Grant No. SC15-R0010, https://www.a-star.edu.sg/) to S.R. and grants from the Ministry of Education, Culture, Sports, Science, and Technology (MEXT) of Japan (Grant No.17H01435, https://www.mext.go.jp/en/index. htm) to H.H.,(Grant No. 17K15123) to T.I. and from Core Research for Evolutional Science and Technology (CREST) of the Japan Science and Technology Agency (Grant No. JPMJCR13W5, https://www.jst.go.jp/kisoken/crest/en/index.html) to H.H., grants from Japan Society for Promotion of Science (Grant No. 17H05895, https://www.jsps. go.jp/english/e-grants/index.html) to T.I. The funders had no role in study design, data collection and analysis, decision to publish, or preparation of the manuscript.

**Competing interests:** The authors have declared that no competing interests exist.

cells, and exhibit planar beating with effective and recovery strokes. On the other hand, 9 +0 motile cilia are found in the embryonic node, show rotational movement and are involved in establishing left-right asymmetry of visceral organs. However, it is not well understood how these two types of motile cilia exhibit their characteristic motion patterns. We have uncovered distinct roles and subcellular localization of the CFAP53 protein in 9+0 versus the 9+2 motile cilia of the mouse. Our data provide novel insights into the molecular basis of motility differences that characterize these two types of mammalian motile cilia.

## Introduction

Cilia are widely found in diverse animals, ranging from protozoans to man. Many of them are immotile, but some of them, such as those found in the mammalian trachea, brain ventricle, reproductive organs and male gametes are motile [1]. Motile cilia are also found within the left-right organizer (LRO) of embryos of many vertebrate species (for example, the ventral node in the mouse embryo), and are required for the initial step of left-right (L-R) patterning [2,3]. Loss of ciliary motility in humans results in congenital disorders such as primary ciliary dyskinesia (PCD), *situs inversus* and heterotaxy. PCD patients exhibit various symptoms including chronic respiratory disease, hydrocephalus, infertility as well as laterality defects [4].

Axonemal dyneins, motor proteins that directly generate the force for ciliary motility, form multisubunit ATPase complexes as outer and inner dynein arms (ODAs and IDAs, respectively) attached to the peripheral doublet microtubules of the cilium. Additional accessory structures, such as radial spokes and nexin links, are involved in the regulation of dynein activity. ODAs, which are absolutely essential for ciliary motility, are formed through several steps. First, dynein proteins synthesized in the cell are pre-assembled into multisubunit complexes in the cytoplasm by several dynein assembly factors (DNAAFs) [5–10]. Assembled complexes are then transported to the basal body, and into the axoneme by intraflagellar transport (IFT), and finally docked to microtubules through the docking complex (DC) composed of TTC25 [11], CCDC114 [12], CCDC151 [13] and ARMC4 [14]. Although many of the genes involved in these steps have been discovered by studying various model organisms and through the identification of causative mutations in human ciliary diseases, the full mechanism remains far from being completely understood.

Even more intriguing is the fact that cilia in different tissues exhibit different patterns of motion. This is best exemplified, for instance, by differences in the beating patterns of cilia within the ventral node versus those that decorate the lumen of the trachea. Monocilia on pit cells of the node beat with a rotary pattern and produce a laminar flow of extraembryonic fluid within the node cavity, while multiple cilia on multiciliated cells (MCCs) of the trachea exhibit a planar back-and-forth beating pattern to clear more viscous mucus from the airways [15]. How these cilia acquire the ability to beat in such distinct patterns has so far been examined only to a limited degree. Node cilia lack the central pair of singlet microtubules and associated radial spoke complex (hence classified as 9+0 cilia), features that are a hall mark of prototypical 9+2 motile cilia, and this ultrastructural modification has been postulated to be the basis for their characteristic rotary beating. This view emanates from genetic evidence from PCD patients as well as zebrafish and mice with mutations in genes that encode proteins of the central pair apparatus (CPA) [16–18]. In all of these situations, while motile cilia-dependent functions in many tissues and organs were compromised, however, laterality was not affected. Moreover, destabilization of the CPA in otherwise 9+2 cilia, such as the tracheal cilia of the

respiratory tract, can cause them to now adopt a rotary beat pattern [17]. Yet another emerging view explaining the causal basis of motility differences between the different kinds of cilia is founded on unique composition of axonemal dyneins in the different cilia-types. Thus, a comprehensive study of axonemal dynein gene expression in the zebrafish has revealed that different motile ciliated tissues express different combinations of axonemal dynein genes [19]. An additional layer of complexity is apparent from the observation that a specific dynein protein can be differentially required for motility in different cilia-types: for instance, mutation of DNAH11 in mice and humans strongly affected nodal cilia motility, but other kinds of motile cilia were far less severely affected [20–22]. All of these data suggest that while different kinds of motility are essential for motile cilia to perform specific kinds of functions, the molecular mechanisms underlying this important aspect of ciliary biology requires further investigation.

We and others have previously identified the coiled-coil containing protein CFAP53 (aka CCDC11) as an important regulator of ciliary motility [22–25]. Loss-of-function studies in the zebrafish and human *situs inversus*/heterotaxy patients have shown that while CFAP53 is essential for cilia motility within the LRO, cilia in other tissues are only mildly affected, similar to the loss of DNAH11 but the converse of mutations affecting the CPA. Using over-expression, we also demonstrated that CFAP53 localized at the base of 9+0 cilia within Kupffer's vesicle (KV, the zebrafish LRO), whereas in 9+2 cilia of the pronephric (kidney) ducts it was present at the base and also along the axoneme [23]. We also documented the pan-axonemal localization of endogenous CFAP53 in multiple cilia of human respiratory MCCs. Furthermore, consistent with motility defects, we also showed that the numbers of ODAs were strongly reduced in zebrafish KV cilia. Despite these intriguing findings implicating a differential requirement of CFAP53 in regulating motility of 9+0 versus 9+2 cilia, the mechanism of this function has remained unexplored.

Here, we have used genetic, cell biological and biochemical analyses to investigate the molecular details of CFAP53 activity in regulating motility in different cilia-types of the mouse. Our study establishes that in addition to the absence or presence of the CPA, differential localization of CFAP53 to 9+0 and 9+2 cilia and its differential effects on the transport and axonemal docking of ODA dyneins is an important contributory factor in the regulation of motility patterns that characterize these cilia-types.

## Results

### Mouse *Cfap53* is specifically expressed in cells with motile cilia

We first examined the expression pattern of the *Cfap53* gene in the mouse using a transgenic strain harboring *Cfap53^lacZ^* bacterial artificial chromosome (BAC), in which *lacZ* has been knocked into the *Cfap53* gene. In embryonic day (E) 8.0 embryos, *Cfap53* was specifically expressed in the node, in particular within the pit cells that are located at the center of the node and have rotating monocilia on their apical surface (Fig 1A and 1B). In the adult mouse, we found *lacZ* expression in epithelial cells of the trachea, brain ventricles, the oviducts and in the testes (Fig 1C–1L), all of which are known to differentiate motile cilia. Thus, *Cfap53* expression in the embryonic and adult mouse is strictly confined to cells with motile cilia.

### Differential requirement of CFAP53 for motility of 9+0 and 9+2 cilia

To uncover the function of CFAP53 in mouse development and physiology, we generated two types of mutants that lack either exon 2 or exons 2–8 of the *Cfap53* gene (Fig 2A). Deletion of exon 2 alone is predicted to produce a severely truncated protein. Since *Cfap53^Δex2/Δex2^* and *Cfap53^Δex2-8/Δex2-8^* mice exhibited indistinguishable phenotypes, the mutant allele lacking exon 2 is referred to as *Cfap53^-/-^* hereafter. *Cfap53^-/-^* mice were smaller at birth and showed laterality

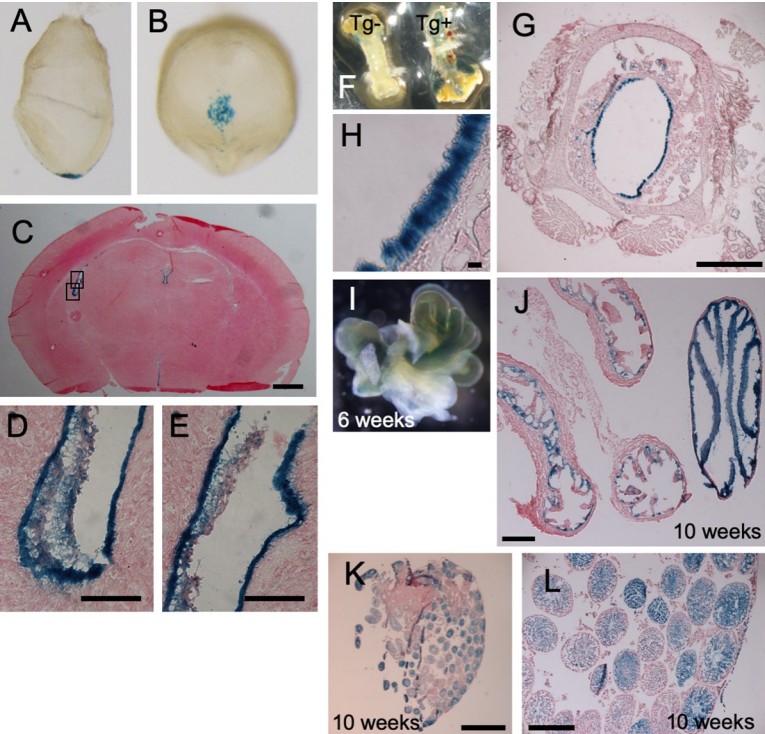

**Fig 1. *Cfap53* is specifically expressed in cells with motile cilia.** *Cfap53^lacZ* mouse embryos and adult tissues were stained with the β-galactosidase substrate X-gal. The images include lateral (A) and ventral (B) views of E8.0 embryos; a sagittal section of the brain at 3 weeks of age (C), with higher magnification views of the boxed regions being presented in (D) and (E); the trachea at 8 weeks of age (F), with an image for a non-transgenic (Tg-) also being shown; sagittal section of the trachea at 8 weeks of age (G), with a higher magnification view shown in (H); the fallopian tube at 6 weeks of age (I); a sagittal section of the fallopian tube (J); and ducts of the testis at 10 weeks of age (K), with a higher magnification view shown in (L). Scale bars, 2 mm (K), 500 μm (C, G, J, L), 50 μm (D, E), 20 μm (H).

defects (Fig 2B and 2C). Half of *Cfap53^-/-* mice (7/14) exhibited *situs inversus*, while 15% (2/14) showed heterotaxy. In addition, all of the *Cfap53^-/-* mice examined (14/14) developed hydrocephalus around 3~4 weeks after birth and died by 6 weeks of postnatal life (Fig 2B and 2D).

Laterality defects observed at newborn stages prompted us to examine several markers of L-R development at E8.0. Whereas *Nodal* is expressed bilaterally at the node and asymmetrically on the left lateral plate mesoderm (LPM) of wild-type (WT) embryos; in *Cfap53^-/-* embryos, however, *Nodal* expression at the node was maintained but that in the LPM was absent (3/3 embryos) (Fig 2E). Since these defects are likely to arise from impairment in the differentiation or motility of the node cilia, we examined these cilia as well as other kinds of motile cilia in the mutant mice. Node cilia are the only motile cilia found in WT embryos at E8.0, and they were completely immotile in *Cfap53^-/-* embryos (Fig 2F and S1 and S2 Videos). The length of node cilia in *Cfap53^-/-* embryos was slightly shorter than that in WT embryos (S1 Fig). Cilia within the trachea and those within the brain ventricles differentiated normally in the absence of CFAP53 and maintained the normal ciliary length (S1 Fig), but in contrast to the nodal cilia, retained motility, albeit with abnormal beating pattern. In particular, the beating angle of the tracheal cilia was smaller, making the effective stroke inefficient (Fig 2G and S3 and S4 Videos). In addition, the beating frequency of the mutant tracheal cilia was significantly reduced (Fig 2H). Cilia within the brain ventricles of the mutants also showed similar defects in their beating pattern, with the proximal portion of the ependymal cilia appearing rigid and with a reduced beating angle and inefficient stroke (Fig 2G), which is consistent with

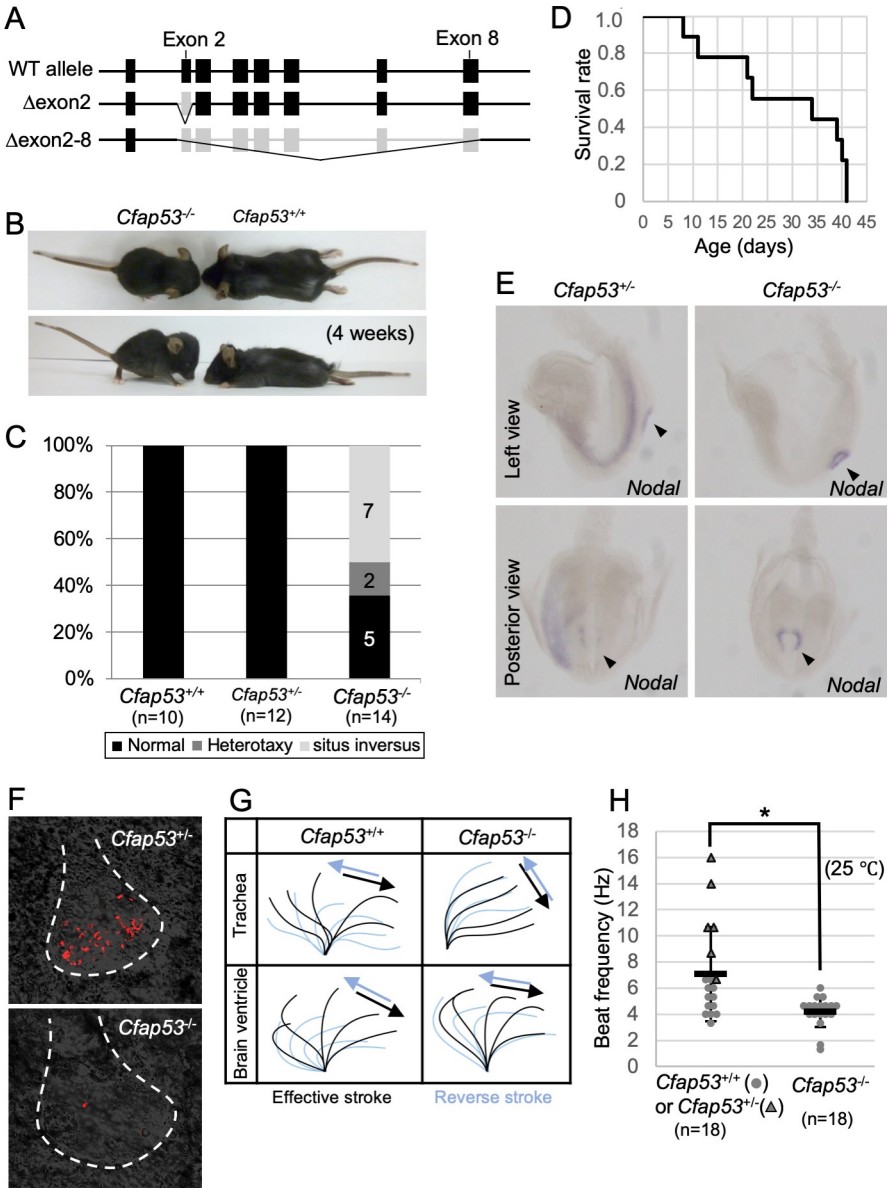

**Fig 2. Laterality defects, hydrocephalus, and ciliary motion defects in *Cfap53* mutant mice.** (A) Genetic structure of the WT mouse *Cfap53* locus and the generation of two types of knockout alleles lacking either exon 2 or exons 2 to 8. (B) Smaller body size and development of hydrocephalus in *Cfap53*$^{-/-}$ mice at 4 weeks of age. (C) Laterality defects of *Cfap53*$^{-/-}$ mice. (D) Survival curve for *Cfap53*$^{-/-}$ mice (n = 14), with all animals dying by 6 weeks of age. (E) *In situ* hybridization analysis of *Nodal* expression in *Cfap53*$^{+/-}$ and *Cfap53*$^{-/-}$ mice at E8.0. *Nodal* expression was missing in the LPM of *Cfap53*$^{-/-}$ embryos. Arrowheads indicate the node. (F) Immotility of node cilia in *Cfap53*$^{-/-}$ embryos at E8.0. Red signals for the *Cfap53*$^{+/-}$ embryo indicate motion trajectory in the corresponding movie. The red signal for the *Cfap53*$^{-/-}$ embryo reflects the trajectory of a bubble. Dashed lines indicate the outline of the node. (G) Wave forms of tracheal and brain ventricle cilia determined from corresponding videos, S3 and S4 Videos. (H) Beat frequency for tracheal cilia of *Cfap53*$^{-/-}$ and control mice (dots and triangles indicate *Cfap53*$^{+/+}$ and *Cfap53*$^{+/-}$ mice, respectively) determined at 25˚C. Data are presented as mean ± SD (n = 18 independent experiments); two tailed Student's t-test (*p = 0.0041). See also S3 Video.

the observation that *Cfap53*$^{-/-}$ mice develop hydrocephalus (Fig 2B). *Cfap53*$^{-/-}$ males were infertile. Their sperm were morphologically abnormal lacking the tails (S2 Fig) and completely immotile (S5 and S6 Videos).

## Localization of CFAP53 in node and tracheal cilia

We investigated the intracellular localization of CFAP53 in mouse tissues with an anti-CFAP53 antibody. In E8.0 embryos, CFAP53 protein was specifically detected at the node of *Cfap53*[+/+] embryos (Fig 3A–3C), but was absent from the node of *Cfap53*[-/-] embryos (Fig 3D–3F). It was highly enriched at the base of the node cilia, most likely at the centriolar satellites [26], although much weaker signal could be detected along the axoneme (Fig 3G). Consistent with this observation, CFAP53 has been reported to be a novel centriolar satellites protein [25]. In fact, CFAP53-positive region at the base of node cilia merged with the centriolar satellites marked by PCM1 (Fig 3G–3I) and surrounded the basal body positive for γ-Tubulin (Fig 3J–3L).

In adult *Cfap53*[+/+] mice, CFAP53 protein was detected in MCCs of the trachea (Fig 3M–3O), but was absent from MCCs of *Cfap53*[-/-] mice (Fig 3P–3R). CFAP53 was localized not only at the base of the tracheal cilia but also in the axonemes (Fig 3M–3O). Overall, CFAP53 localized mainly at the base (centriolar satellites) of node cilia, whereas it was also present prominently in the axonemes of tracheal cilia.

We also investigated intracellular localization of CFAP53 protein by establishing a transgenic strain harboring a *Cfap53*[Venus] BAC, encoding functional CFAP53::Venus fusion protein (S3 Fig). Like the endogenous CFAP53 protein, the CFAP53::Venus fusion protein was expressed at the node, and was enriched at the base of node cilia although much weaker Venus signal was detected in the axoneme (S4 Fig). In the tracheal cilia of the transgenic mice, the fusion protein was detected at the base of MCC cilia and in their axonemes (S4 Fig). Unlike the endogenous CFAP53 protein, however, the fusion protein was absent in the distal end of the axonemes of tracheal cilia of *Cfap53*[Venus] BAC, *Cfap53*[-/-] mice (S4 Fig), suggesting that the CFAP53::Venus protein may fail to be transported efficiently to the distal region of tracheal cilia.

## ODAs are lost from node cilia but mostly retained in tracheal cilia of *Cfap53*[-/-] mice

Given that node cilia lose motility in the absence of CFAP53, we next examined the presence or absence of DNAH5, a major axonemal dynein heavy chain protein, by immunostaining. In WT node cilia, DNAH5 is localized throughout the axoneme (Fig 4A–4C). In *Cfap53* mutant node cilia, however, DNAH5 was not detected in the axoneme (Fig 4D–4F). In tracheal cilia, on the other hand, DNAH5 was detected throughout the axoneme, in both control (Fig 4G–4I) and in *Cfap53*[-/-] mice (Fig 4J–4L). Since DNAH5 is essential for ciliary motility [27,28], the absence or presence of DNAH5 in the two types of cilia correlated with effects of the loss of CFAP53 on their ability to move.

We also examined the localization of two other important ODA dynein heavy chains, DNAH11 and DNAH9. Since antibodies suitable for immunostaining of DNAH11 in mouse embryos are not available, we generated the *Dnah11*[Venus] allele, which encodes a functional DNAH11::Venus fusion protein (S5 Fig), and examined its localization. DNAH11 localized along the entire axoneme of node cilia (Fig 5A–5C) as previously described [11,29], while it localized only to the proximal portion of trachea cilia (Fig 5G–5I), as reported earlier from studies with human respiratory MCCs [11]. However, DNAH11 was absent from node cilia of *Cfap53*[-/-] embryos (Fig 5D–5F). By contrast, proximal localization of DNAH11 in tracheal cilia was maintained in *Cfap53*[-/-] mice (Fig 5J–5L).

In humans, DNAH9 is localized to the distal region of tracheal cilia, and in PCD patients with *DNAH9* mutations, the motility of the tracheal cilia is affected. Many of these patients also exhibit *situs* defects implying a requirement for the DNAH9 protein in human nodal cilia

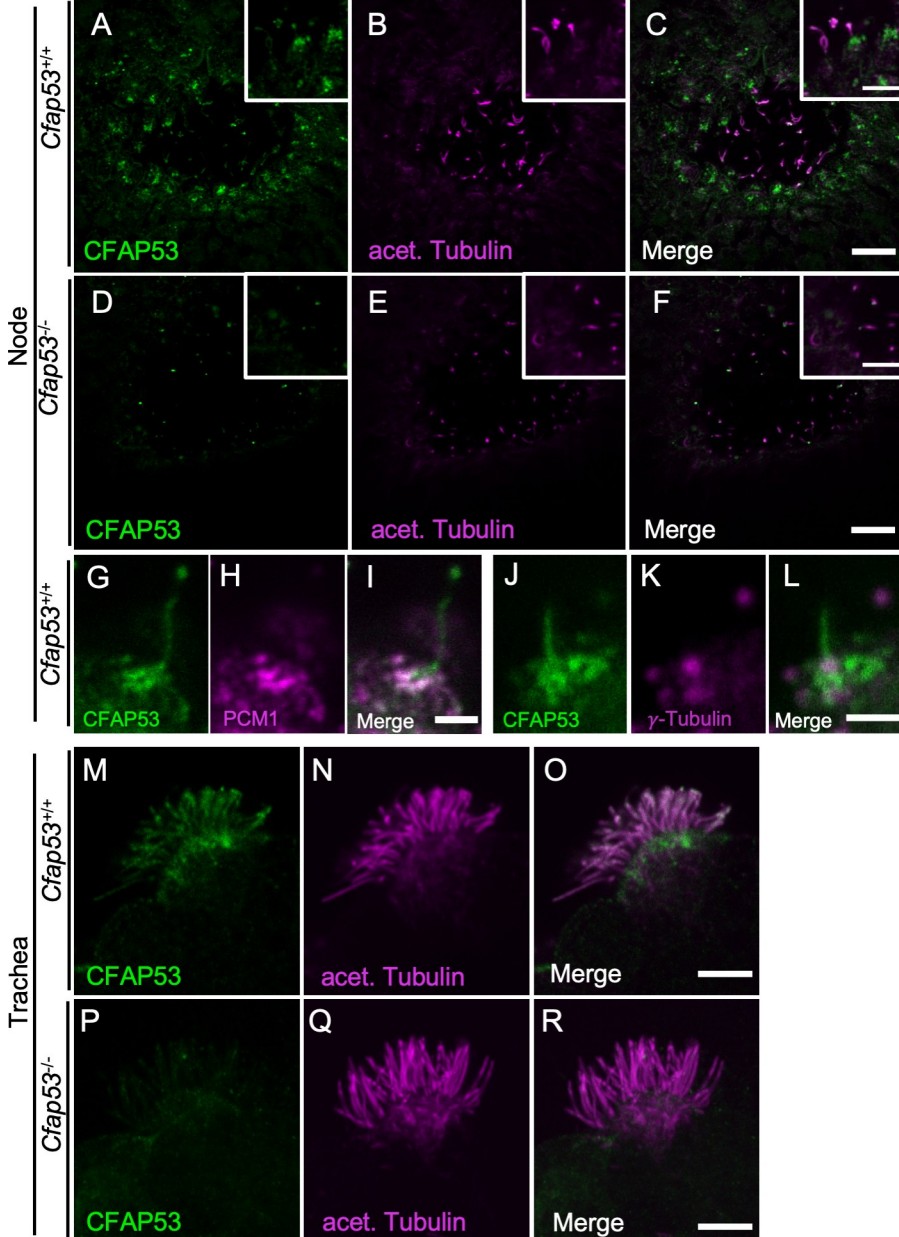

**Fig 3. Differential localization of CFAP53 in tracheal and node cilia.** (A–L) Immunofluorescence staining of the node of WT (A-C, G-L) or *Cfap53*$^{-/-}$(D-F) embryos at E8.0 with antibodies to CFAP53 (A, D, G, J), to acetylated (acet.) Tubulin (B and E), to PCM1 (H) and to γ-Tubulin (K). Scale bars, 10 μm (A-F). Higher magnification images are shown in insets. Scale bars, 5 μm (A-F). Note that CFAP53 is mainly localized at the base of nodal cilia (centriolar satellites positive for PCM1), and weakly in the axoneme. Scale bar, 2 μm (G-L). (M-R) Immunofluorescence analysis of an isolated tracheal MCC from a WT (M-O) or *Cfap53*$^{-/-}$(P-R) mouse. CFAP53 is detected at the base of cilia (the centriolar satellites) and in the axoneme. Scale bar, 5 μm.

motility [30]. In tracheal cilia of *Cfap53*$^{-/-}$ mice, distal localization of DNAH9 was maintained, although its level was reduced compared with that apparent in control mice (S6 Fig). Unexpectedly, we failed to detect DNAH9 protein in node cilia, either with a DNAH9-specific antibody (S6 Fig) or by live imaging of a DNAH9::Venus fusion protein encoded by a *Dnah9*$^{Venus}$ transgene (S7 Fig). To confirm the absence of DNAH9 at the node of the mouse embryo, we

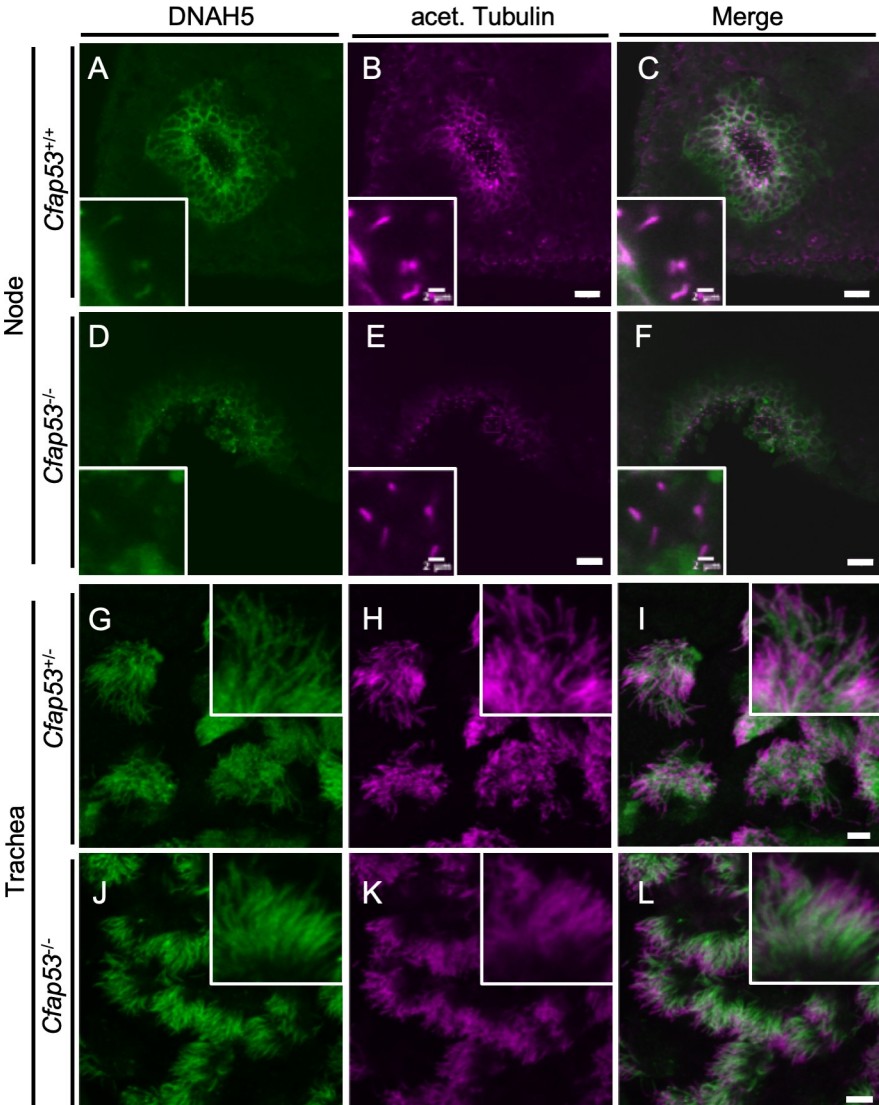

**Fig 4. Localization of DNAH5 in ciliated cells of the trachea and node.** (A-F) Immunofluorescence staining with antibodies to DNAH5 (green) and to acetylated (acet.) Tubulin (magenta) of the node at E8.0 and adult trachea (G-L) from *Cfap53*−/−and control mice. (D-F, J-L) DNAH5 was absent from node cilia but maintained in tracheal cilia of *Cfap53*−/−mice. Scale bars, 20 μm. Insets show higher magnification views. Scale bars, 2 μm.

generated a mutant mouse deficient in this protein (*Dnah9* KO mouse; Fig 6A). *Dnah9* KO mouse showed normal laterality (9/9 newborn mice; Fig 6B–6D), unlike in humans where *DNAH9* mutations result in laterality defects [30]. When *Dnah9* KO mice were dissected, L-R asymmetries of the lung, the heart apex, great arteries, azygous vein, stomach, liver and spleen were found to be normal (Fig 6D). Around one month after birth, four of the six *Dnah9* KO mice examined showed hydrocephalus (Fig 6E). Tracheal and ependymal cilia from *Dnah9* KO mice were motile, but exhibited abnormal beat pattern with reduced beating angle and insufficient effective stroke (Fig 6F, S7–S10 Videos). The beat frequency of mutant tracheal cilia was reduced (Fig 6G). Sperm from a *Dnah9* KO male was morphologically normal and retained normal motility (S11 and S12 Videos), which is consistent with the previous report that DNAH9 is not expressed in mouse sperm [31].

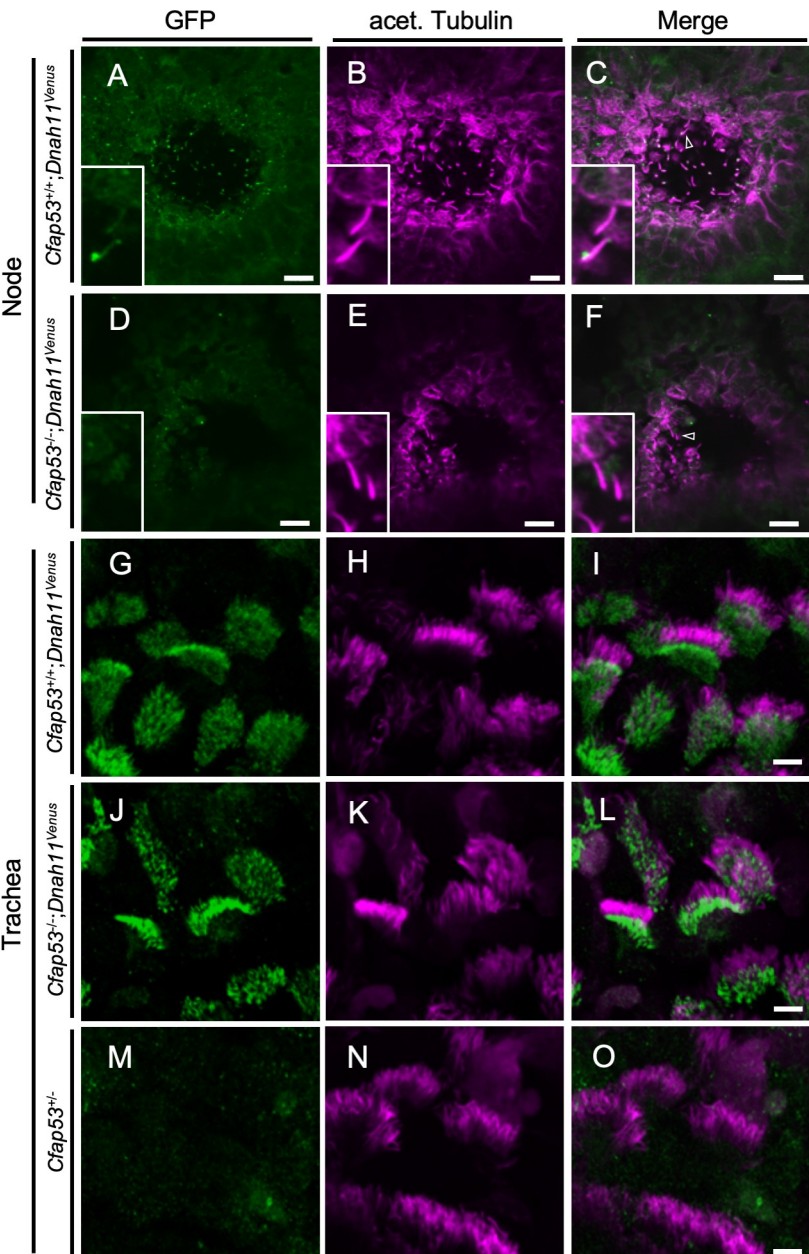

**Fig 5. Localization of DNAH11 in ciliated cells of the trachea and node.** (A-F) Immunofluorescence staining with antibodies to GFP (green) and to acetylated (acet.) Tubulin (magenta) for the node at E8.0 and adult trachea (G-O) of *Cfap53$^{-/-}$; Dnah11$^{Venus}$* and control mice. (D-F, J-L) DNAH11::Venus was absent from node cilia but was maintained in tracheal cilia of *Cfap53$^{-/-}$* mice. In (C) and (F), the arrowhead indicates a node cilium. Scale bars, 10 μm (A-F) or 5 μm (G-O). Insets represent higher magnification views.

We next studied the ultrastructure of node and tracheal cilia of WT and *Cfap53* mutants using transmission electron microscopy (TEM). The axonemes of node cilia in WT embryos have a 9+0 configuration with ODAs attached to the nine peripheral microtubule doublets (Fig 7A and 7B). However, in node cilia from *Cfap53$^{-/-}$* embryos, ODAs were missing from most of the sections examined (Fig 7C and 7D and 7J). Tracheal cilia from WT mice show 9+2 configuration with the CPA and ODAs attached to the peripheral microtubule doublets (Fig

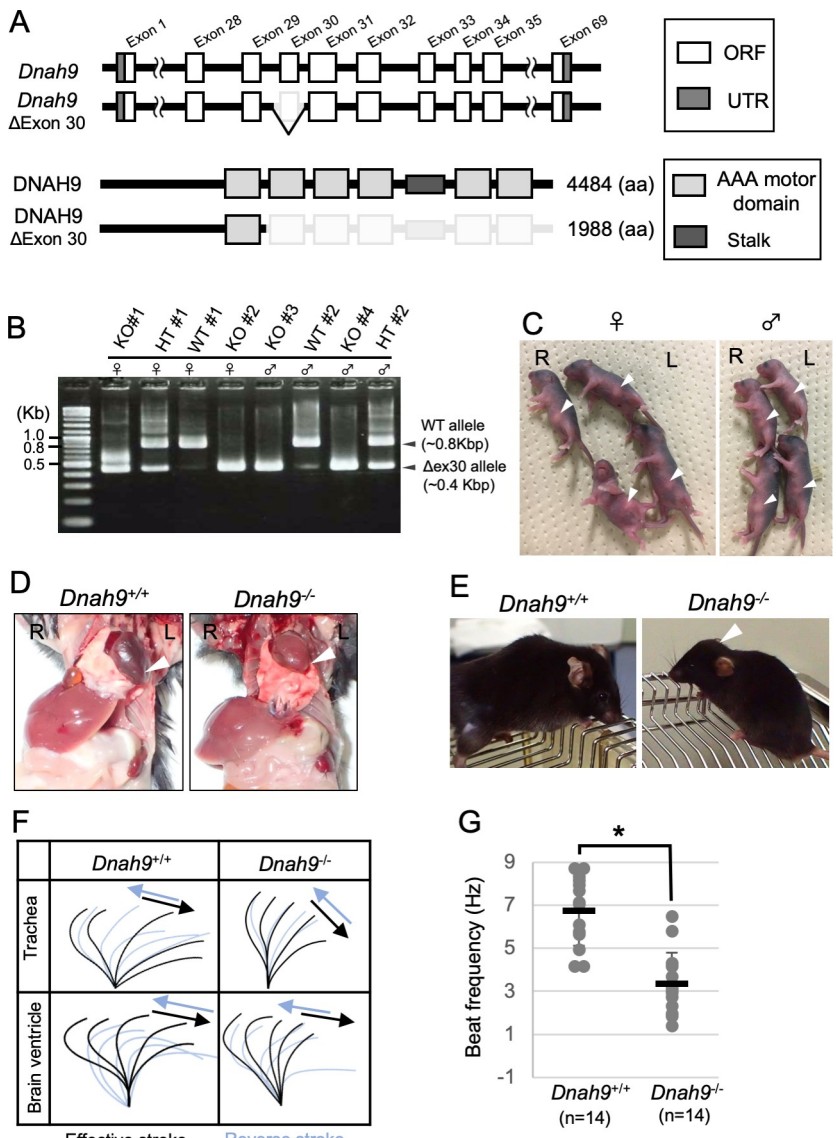

**Fig 6. Phenotype of *Dnah9* mutant mouse.** (A) The exon-intron structure of mouse *Dnah9* gene and generation of knockout allele lacking exon 30. Note that the motor domains of DNAH9 are encoded by exons 29–32 and exons 34–35. (B) Eight pups generated by CRISPR-mediated mutagenesis were genotyped by PCR. (C) External appearance of the same eight pups whose genotype is shown in (B). Arrowheads denote the position of the stomach. Note that all the *Dnah9*−/− (KO) mice have the stomach at the normal position. (D) Laterality of visceral organs in WT and *Dnah9*−/− mice. The arrowheads indicate the position of the heart apex. (E) *Dnah9*−/− mice developed hydrocephalus (arrowhead). (F) Wave forms of tracheal and brain ventricle cilia of *Dnah9*+/+ and *Dnah9*−/− mice. See also S7–S10 Videos. (G) Beat frequency for tracheal cilia of *Dnah9*−/− and control mice. Data are presented as mean ± SD (n = 14 independent experiments); two tailed Student's t-test (*p<0.05). See also S7 and S8 Videos.

7E and 7F). In tracheal cilia of *Cfap53*−/− mice, most ODAs were retained (Fig 7G, 7I and 7K), consistent with our observations that these cilia of the mutants are able to move, albeit with an altered motility pattern (Fig 2G). When tracheal cilia were examined along the proximal-distal direction, there was no significant preference in the loss of ODAs (S8 Fig). Thus, the partial loss of ODAs from tracheal cilia (with most axonemes having five to eight ODAs) allowed them to maintain their motility, but impaired their motion pattern.

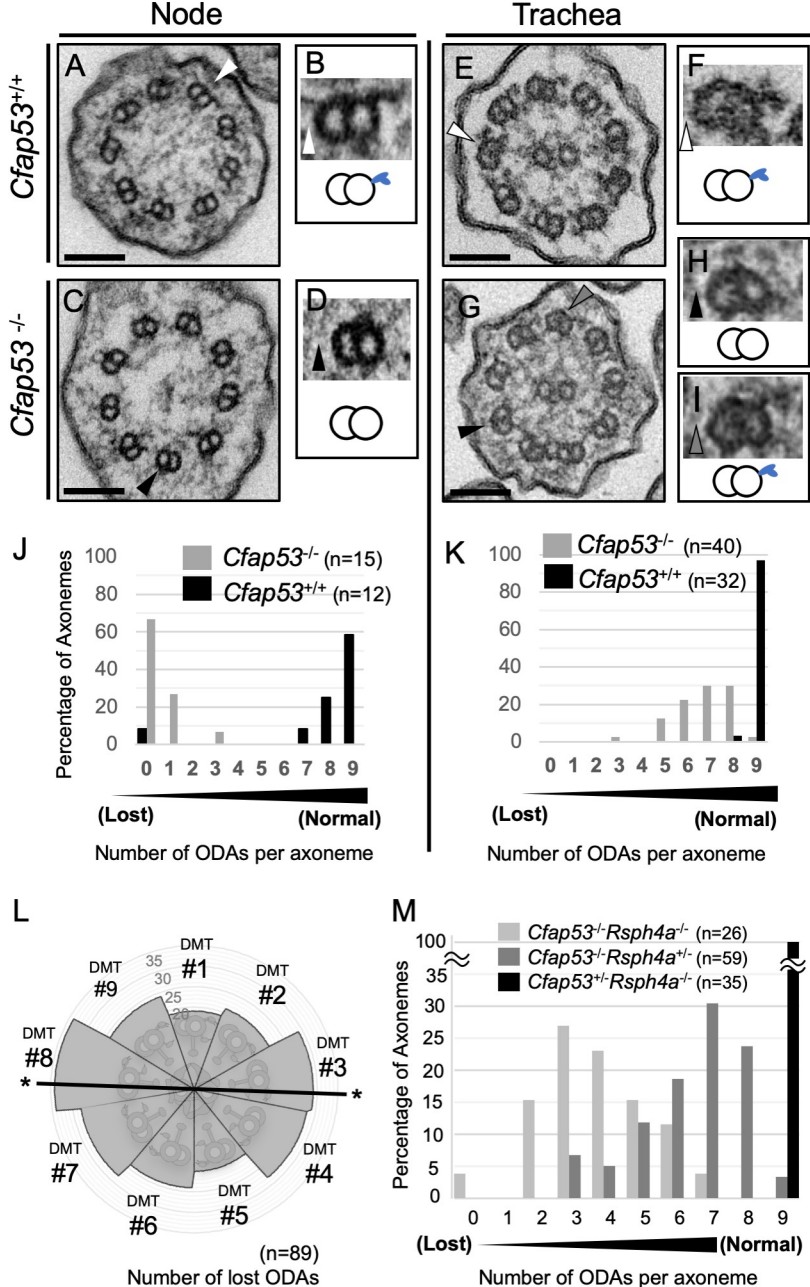

**Fig 7. ODAs are completely lost in node cilia but largely maintained in tracheal cilia of *Cfap53*−/−mice.** (A–D) TEM of node cilia of *Cfap53*+/+ (A) and *Cfap53*−/− (C) embryos at E8.0. Higher magnification views of corresponding doublet microtubules indicated by the arrowheads are shown in (B) and (D), respectively, together with schematic diagrams of the microtubule pairs and ODAs (blue protrusions). Scale bars, 100 nm. (E–I) TEM of tracheal cilia of adult *Cfap53*+/+ (E) and *Cfap53*−/−(G) mice. Higher magnification views of corresponding doublet microtubules indicated by the arrowheads are shown in (F), (H), and (I) together with schematic diagrams of the microtubule pairs and ODAs. Scale bars, 100 nm. (J and K) Distribution of the number of ODAs per axoneme for node (J) and tracheal (K) cilia of *Cfap53*−/−and *Cfap53*+/+ mice determined from TEM images. (L) Nine peripheral doublet microtubules were numbered from DMT1-DMT9, on the base of their relative position to the central pair: DMT1 is the doublet microtubule located at the position perpendicular to the central pair microtubules. The frequency of ODA loss at each DMT of *Cfap53*−/−tracheal cilia is shown by a circular plot. Note that DMTs with the line indicated by asterisk (#3 and #8) are preferentially lost ($\rho$ = 0.04263 by Rayleigh's test). (M) Distribution of the number of ODAs per axoneme for tracheal cilia of *Cfap53*; *Rsph4a* double mutant mice determined from TEM images.

Interestingly, ODAs at the position 3 and 8 were preferentially lost in tracheal cilia of *Cfap53*[-/-] mice ([Fig 7L]). It may be that ODAs are weakly attached to microtubules in the absence of CFAP53, and ODAs at position 3 and 8 are preferentially detached from microtubules because planar beating likely takes place along the ODA 3–8 axis. To test this, we examined the tracheal cilia of *Cfap53*[-/-], *Rsph4a*[-/-] double mutant (DKO) mice, since tracheal cilia exhibit rotational movement instead of planar beating in the absence of RSPH4a [17]. ODAs in tracheal cilia of *Cfap53*[-/-]; *Rsph4a*[-/-] DKO mouse were more severely lost than those of *Cfap53*[-/-] mouse ([Fig 7M]).

## CFAP53 associates with axonemal microtubules of tracheal cilia and interacts with dynein proteins and the docking complex (DC)

Given that CFAP53 is an axoneme-associated protein in tracheal cilia, we investigated its localization within these cilia more precisely. First, we examined whether CFAP53 interacts with microtubules. Isolated tracheal cilia were treated with high salt (0.6 M NaCl or NaI), and then subjected to centrifugation to obtain soluble and insoluble fractions. Immunoblot analysis revealed that, after treatment with 0.6M NaCl, microtubules remained insoluble, with tubulin being detected in the precipitate fraction ([Fig 8A]). However, the dynein complex became detached from the microtubules, with ODA dynein intermediate chain 2 (DNAIC2) being detected in the soluble fraction. Under this condition, CFAP53 was present in the precipitate, suggesting that CFAP53 protein remained associated with microtubules. When the tracheal cilia were treated with 0.6M NaI, which disrupts most protein-protein interactions, microtubules were dissociated and tubulin was detected in the soluble fraction. Under this condition, DNAIC2 and CFAP53 were also found largely in the soluble fraction ([Fig 8A]). These results suggest that CFAP53 is tightly associated with microtubules in tracheal cilia.

The tight association of CFAP53 protein with microtubules suggests that CFAP53 may interact with ODA dyneins and/or DC component(s). We therefore examined potential physical interaction of CFAP53 with three ODA dynein components (the dynein intermediate chains DNAIC1, DNAIC2 and the heavy chain DNAH11) and three components of the DC (TTC25, CCDC114 and CCDC151) by co-immunoprecipitation analysis of transfected HEK293T cells (Figs [8B–8E], [S9]). TTC25 is thought to be the DC component most proximal to the axonemal microtubules [32], while CCDC151 is the most distal DC component [13]. Over-expressed CFAP53 interacted with over-expressed DNAIC1, DNAIC2, DNAH11 and TTC25, but not with CCDC114 or CCDC151 ([Fig 8B–8E], [S9]). These results suggest that in the axoneme of tracheal cilia, CFAP53 is embedded in microtubules and may directly interact with the DC via TTC25. CFAP53 at the centriolar satellites of node and tracheal cilia may also interact with TTC25 and ODA proteins, facilitating their transport into the axoneme.

## Differential requirement of CFAP53 for axonemal localization of TTC25 in the node and tracheal cilia

Given that CFAP53 physically interacts with TTC25, we next examined the genetic relationship between CFAP53 and TTC25. In *Ttc25*[-/-] mutant mice [32], localization of CFAP53 was maintained in node cilia ([Fig 9A–9F]), suggesting that the correct localization of CFAP53 does not require TTC25. In contrast, the subcellular localization of TTC25 was dependent on CFAP53, as revealed by the observation that in the node cilia of *Cfap53*[-/-] embryos, TTC25 protein was absent from the axonemes ([Fig 9G–9L]). Given that TTC25 is a DC component located most proximally to the axonemal microtubules and mutation of TTC25 in mice and

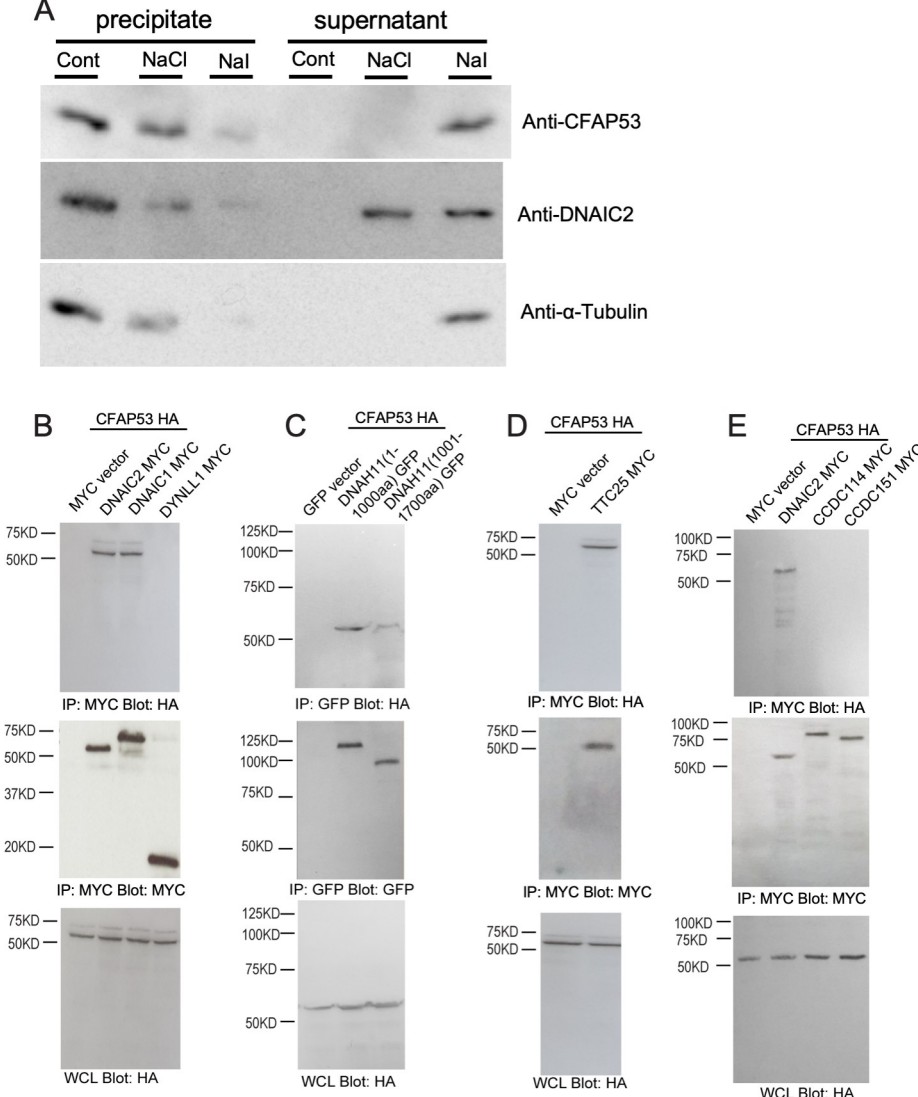

**Fig 8. Association of CFAP53 with axonemal microtubules of tracheal cilia as well as with DNAIC proteins and TTC25.** (A) Isolated axonemes of tracheal cilia were treated (or not, Cont) with 0.6 M NaCl or NaI and then centrifuged at $15000 \times g$ for 12 min at 4˚C, and the resulting precipitate and supernatant fractions were subjected to immunoblot analysis with antibodies to CFAP53, to DNAIC2 (ODA marker) and to α-Tubulin. (B–E) HEK293T cells transfected with expression vectors for HA-tagged CFAP53 or MYC epitope-tagged DNAIC1, DNAIC2, DYNLL1 (cytoplasmic dynein light chain), TTC25, CCDC114, or CCDC151 (or with the corresponding empty vectors), or GFP-tagged DNAH11 N-terminus fragments 1–1000 aa or 1001–1700 aa, as indicated, were subjected to immunoprecipitation (IP) with antibodies to HA or to MYC or to GFP, and the resulting precipitates as well as the original whole cell lysates (WCLs) were subjected to immunoblot analysis with antibodies to HA, MYC or GFP. CFAP53 interacted specifically with the axonemal dyneins DNAIC1, DNAIC2 and DNAH11 N-terminus and the DC member TTC25.

humans inhibits the association of ODAs with axonemes [32], the lack of TTC25 from the axonemes of *Cfap53*⁻/⁻ node cilia is likely to be responsible for their severe lack of ODAs. On the other hand, in tracheal cilia, axonemal localization of TTC25 protein does not strictly require CFAP53 function, and TTC25 was found in the axonemes of tracheal cilia of *Cfap53* mutants, although at reduced levels (S10 Fig). This reduced level of TTC25 likely accounts for the partial loss of ODAs from the tracheal cilia (Fig 7K).

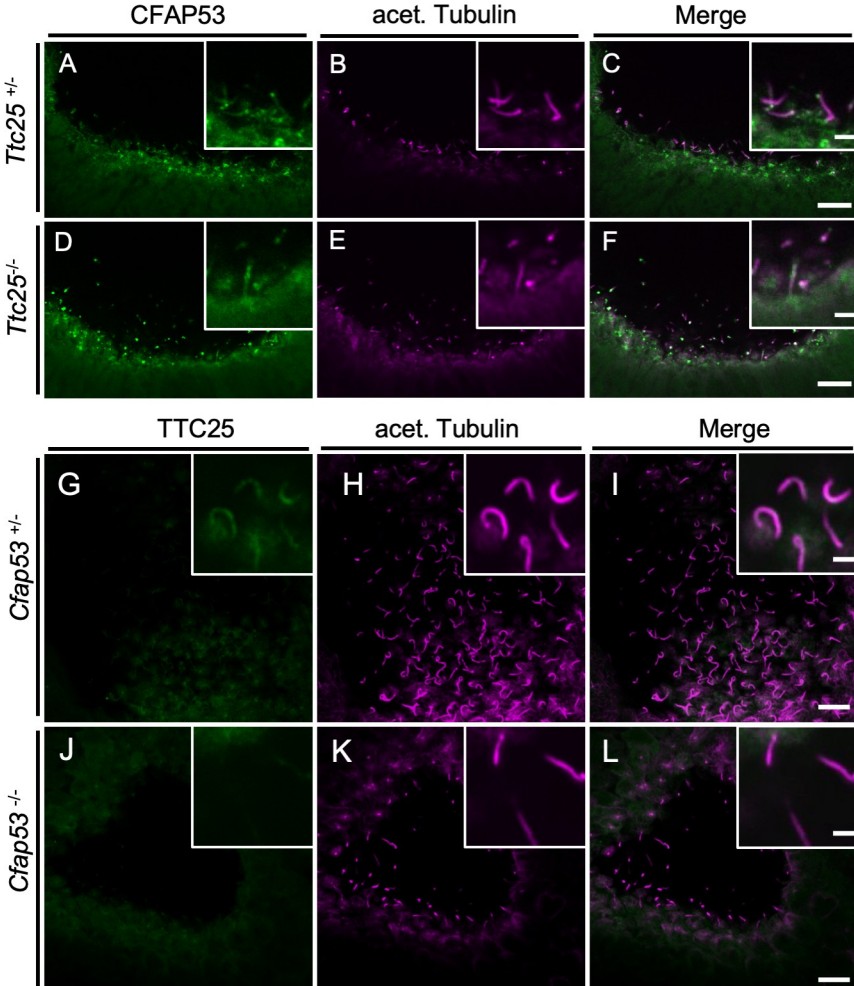

**Fig 9. Genetic interaction between *Cfap53* and *Ttc25* in node cilia.** (A–F) Immunofluorescence staining with antibodies to CFAP53 (green) and to acetylated (acet.) Tubulin (magenta) of the node from *Ttc25*$^{-/-}$ and control embryos at E8.0. CFAP53 was localized at the axoneme and base of node cilia of *Ttc25*$^{-/-}$ embryos. (G–L) Immunofluorescence staining with antibodies to TTC25 (green) and to acetylated (acet.) Tubulin (magenta) of the node from *Cfap53*$^{-/-}$ and control embryos at E8.0. TTC25 was lost from the axoneme of node cilia in the *Cfap53*$^{-/-}$. Scale bars, 10 μm. Insets show higher magnification views. Scale bars, 2 μm.

## Discussion

Motile cilia and flagella are involved in multiple developmental and physiological processes that involve flow of fluids of differing viscosities over epithelial surfaces as well as the locomotory activities of individual cells [15]. In each of these instances, the beating patterns of these organelles is harmonized with the specific kind of biomechanical force that is required for the particular function. In mammals, this is particularly well apparent in the case of ciliary motility in the node for determination of L-R asymmetry, motility of multiple cilia on respiratory MCCs for mucus clearance and movement of flagella for swimming movement of sperm toward the egg during fertilization. Disruption to these specific motility patterns underlie a range of disease phenotypes manifest in PCD patients like *situs* abnormalities, respiratory insufficiency and infertility. Given these considerations, how these unique patterns of ciliary motility, which are matched with specific functions of the motile cilia, are regulated at the molecular level, is a topic of substantial biological and clinical significance.

Previous work from our group and that of Noël et al. have implicated the coiled-coil containing protein CFAP53 in differentially regulating the motility of zebrafish and human cilia [22,23]. Our current in-depth analysis of CFAP53 function in mouse cilia using stable genetic mutations have clearly established that the protein is absolutely essential for the motility of the 9+0 node cilia, but not of the 9+2 motile cilia such as the tracheal and ependymal cilia. Also, our data show that CFAP53 is not essential for the formation of immotile primary cilia or the multiple cilia on MCCs per se, unlike what has been described by others using transient knock-down of the protein in cultured retinal pigmented epithelial (RPE) cells, cultured human MCCs and epidermal MCCs of *Xenopus* larvae [25]. As suggested by others [33], acute and chronic loss of CFAP53 may result in distinct ciliary phenotypes. These and other studies have also described several different subcellular localization patterns for the CFAP53 protein: the centriole and actin cytoskeleton in human skin fibroblasts [34], on ciliary axonemes in zebrafish kidneys and human respiratory cells, at the basal body in zebrafish KV and kidney cells [23] and at the basal body and centriolar satellites in RPE cells and human and frog MCCs [25]. Our data now clarify the expression and localization of CFAP53 *in vivo*: in the mouse, the *Cfap53* gene is specifically expressed in cells with motile cilia and not in those that differentiate primary cilia, and the protein is localized differently between the node (9+0) and tracheal (9+2) cilia, similar to what we have described previously for cilia in zebrafish and human MCCs [23]. Thus, the CFAP53 protein is localized prominently at the base (the centriolar satellites) of the node as well as tracheal cilia, and along the axonemes of the tracheal cilia.

In node cilia, CFAP53 at the centriolar satellites may regulate the transport of TTC25 and ODA dyneins by interacting with these proteins at the ciliary base. This is supported by the absence of TTC25, DNAH5 and DNAH11 from the axonemes of *Cfap53*[-/-] node cilia (Fig 10A). The mechanism by which ODAs are transported to the axoneme remains largely unknown. As mentioned in the introductory section, ODA proteins are preassembled in the cytoplasm into large complexes by several DNAAFs and are transported to the ciliary base. They then enter into the cilium and get transported toward the tip by IFT [35]. Given that centriolar satellites regulate protein trafficking to cilia and ciliary protein content [36,37], CFAP53 at the centriolar satellites may form a complex with TTC25 and ODAs, including DNAH5 and DNAH11, and regulate their trafficking into the cilium (Fig 10B). On the other hand, in tracheal cilia, CFAP53 is localized not only at the base but also within the axoneme. In the axoneme, CFAP53 is tightly associated with microtubules, likely linking ODAs to microtubules via the DC, and participating in stable attachment of ODAs to microtubules. Consistent with this view, a recent study of various microtubule interacting proteins (MIPs) from *Chlamydomonas* has shown that CFAP53 is a MIP that is located within the A tubule of the flagella [38]. Although MIPs have been proposed to confer stability to the 9+0 axoneme [39], the morphology and the number of node cilia remained largely normal in *Cfap53* mutant mice. CFAP53 may also be located at or near the surface of microtubules of mammalian cilia, but verification of this possibility warrants further investigation. Alternatively, physical association of CFAP53 with TTC25 and DNAH11 (Fig 8) may reflect their interaction at the centriolar satellites.

ODAs were lost from node cilia, but were largely maintained in tracheal cilia of *Cfap53*[-/-] mice. However, it remains unclear whether IDAs are affected in 9+2 cilia of *Cfap53*[-/-] mice. Technical difficulties to detect IDAs in vertebrate cilia by TEM and the lack of appropriate antibodies against IDA components prevented us from analyzing this issue. Axonemal localization of DNAH11 and DNAH5 was lost from *Cfap53*[-/-] node cilia, but was largely maintained in *Cfap53*[-/-] trachea cilia (Fig 10A), implying the presence of a CFAP53-independent mechanism that transports TTC25 and ODAs into the axoneme of 9+2 cilia and also participates in their stable docking with the axonemal microtubules (Fig 10B). Apparently, the axonemal structure of 9+2 cilia is more complex than that of 9+0 cilia, in that the former have additional

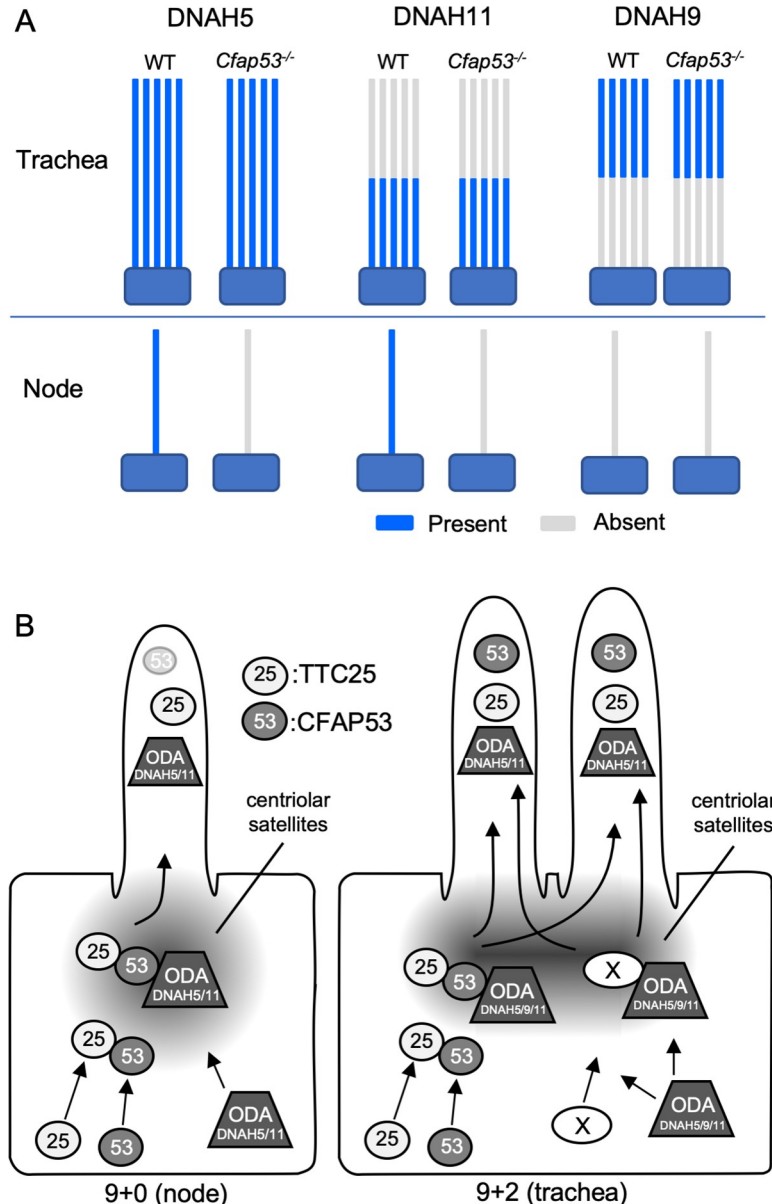

**Fig 10. Differential role of CFAP53 in 9+0 and 9+2 motile cilia.** (A) Schematic diagram for the expression of ODA proteins (DNAH5, DNAH11, DNAH9) in node and tracheal cilia of *Cfap53⁻/⁻* and WT mice. All three dynein heavy chain proteins were maintained in tracheal cilia, whereas DNAH5 and DNAH11 were completely lost in node cilia of the mutant mice. DNAH9 was not detected in node cilia of WT or *Cfap53⁻/⁻* mouse embryos. DNAH5 and DNAH11 (or DNAH9) are indicated by blue color on the axonemes. (B) CFAP53 (53) interacts with TTC25 (25) at the centriolar satellites. It also interacts with ODA proteins including dynein heavy chains such as DNAH11 and possibly DNAH5/9. CFAP53 likely facilitates the transport of TTC25 and the dyneins into cilia. An unknown protein (X) that can also interact with TTC25 and ODA proteins and mediate their transport into the tracheal cilia in the absence of CFAP53 is either not expressed or is nonfunctional in node cells.

structures such as the CPA. This may be the reason why cells with 9+2 cilia could possess more elaborate mechanisms for the transport and docking of axonemal dyneins. We have searched for an axonemal protein gene that is expressed in tracheal ciliated cells but not in the node cells. Examination of the expression of 18 ciliary genes, however, failed to identify such a trachea-specific gene (S10 Fig). Alternatively, like what we have described for CFAP53, a protein

present in both cilia-types but with differential localization and function in each kind of cilia could also fulfil this role. In any case, further investigation is needed to identify the protein that might function at the base of 9+2 cilia to permit transport of DC members and ODAs into the axoneme or to stabilize the attachment of the ODAs to the axonemal microtubules (Fig 10B).

Rather unexpectedly, our study has uncovered different roles of DNAH9 in the motility of the nodal cilia between human versus the mouse. Several patients with mutations in the *DNAH9* gene have been reported to exhibit laterality defects, suggesting that DNAH9 is required for motility of node cilia in man [30]. Also, *Dnah9* mRNA is expressed in the node of the mouse embryo [30]. However, we were unable to detect DNAH9 protein in the mouse node either with an anti-DNAH9 antibody or by imaging the *Dnah9*$^{Venus}$ mouse. By contrast, the protein could be clearly detected in tracheal cilia using these strategies. Furthermore, mice lacking DNAH9 failed to show laterality defects (Fig 6). The absence of DNAH9 protein in node cilia despite the presence of *Dnah9* mRNA at the node is surprising. *Dnah9* mRNA may be subject to translation repression in node cells or the translated protein may fail to be targeted to the axoneme. The differential requirement of DNAH9 in node cilia motility between human and mouse is indeed unprecedented. It has been suggested that DNAH5 forms a heterodimer with DNAH9 or DNAH11, and the presence of either DNAH5-9 or DNAH5-11 heterodimer is sufficient for ciliary motility [11]. In view of these interactions, DNAH9 and DNAH11 may be located in distinct portions of human node cilia as they are in tracheal cilia, and both kinds of heterodimers may be required for smooth rotational movement of the node cilia.

In conclusion, our study of CFAP53 function in the mouse has revealed novel molecular differences in the mechanisms that underlie the distinct patterns of motility between mammalian 9+0 and 9+2 cilia. While the current understanding of the motility differences between these cilia-types has majorly focused on the presence or absence of the CPA, our findings highlight the importance of the basal body, and perhaps the associated centriolar satellites, in the differential regulation of axonemal docking protein and ODA dynein transport into the axoneme. In addition, we have documented that 9+0 and 9+2 cilia have unique ODA composition with respect to DNAH9, which is also likely to be a contributory factor to the motility differences that characterize these two kinds of cilia. Importantly, this evolving mechanistic framework underlying distinct cilia motility patterns will allow the interpretation, at the molecular level, of the causal basis of the phenotypic spectrum exhibited by ciliopathy patients with differential effects on the motility of cilia in different regions of their bodies.

## Materials and methods

### Ethics statement

All mouse experiments were approved by the Institutional Animal Care and Use Committee (Permission number: A2016-01-6) and carried out in accordance with guidelines of RIKEN BDR.

### Mice

Mice were maintained at the Animal Facility of Riken Center for Biosystems Dynamics Research (BDR), Japan, under a 12-h light, 12-h dark cycle and were provided with food and water *ad libitum*. Two types of *Cfap53* knockout mice, lacking exon 2 or exons 2 to 8, were generated with the CRISPR-Cas9 system as performed with guide RNA 1 (located upstream of exon 2 5'-CCTCTTAATTTTTACTTATTGTA-3'), guide RNA 2 (5'-GCTTCAACTCCTGCA CCACCAGG-3') and guide RNA 3 (5'-GGTGAGCCAAAATATGGGCC-3'). For generation

of *Cfap53*<sup>Venus</sup> BAC transgene, the Venus coding sequence was inserted immediately before the stop codon of *Cfap53* in the BAC clone RP23-99F21. For generation of the *Cfap53*<sup>lacZ</sup> transgene, an IRES (internal ribosome entry site)-*lacZ* cassette was inserted immediately after the stop codon of *Cfap53* in the same BAC clone. The *Dnah11*<sup>Venus</sup> allele was generated by inserting Venus into exon 43 of *Dnah11* gene in frame. The *Dnah9*<sup>Venus</sup> allele was generated by inserting Venus at the carboxyl terminus of DNAH9. The construction of the *Ttc25*<sup>-/-</sup> mice has been described previously [32]. *Dnah9* knockout mice, lacking exon 30, were generated with the CRISPR-Cas9 system as performed with guide RNA 1 (located upstream of exon 30 5'-TCAGACATAGAAGTGCGAGATGG-3') and guide RNA 2 (5'-GATCGATCTTACAGC TTGTCTGG-3'). *Cfap53* mutant mice were generated and maintained on a BC57BL background, while *Cfap53::Venus* Bac transgenics were generated by injecting BAC DNA into the pronuclei of (C57BL X C3H) F1 mouse eggs.

### *In situ* hybridization and staining for *lacZ* expression

Whole-mount *in situ* hybridization was performed according to standard protocols with specific RNA probes for *Nodal*. Staining for *lacZ* expression was performed with the β-galactosidase substrate X-gal (5-bromo-4-chloro-3-indolyl-β-D-galactopyranoside) as described previously [40]. Serial sections (thickness of 10 μm) of the brain, trachea, oviduct, and testis were prepared after X-gal staining.

### Antibodies

cDNA for the C-terminal region of mouse CFAP53 was cloned into the bacterial expression vector pMaL-c2X (New England Biolabs) with the use of the forward and reverse primers 5'-AGAGTCTATGCAGCGGGAGT-3' and 5'-TAGAGGAGGCTTGTCAGGGTA-3', respectively. CFAP53 expressed as a fusion protein with maltose binding protein in *Escherichia coli* was purified from the bacterial cells with the use of amylose resin (New England Biolabs) and injected into rabbits for the generation of antibodies. Specific antibodies were purified from rabbit serum by affinity chromatography. Additional antibodies used in this study are rabbit anti-GFP (Invitrogen, A-11122), mouse anti-acetylated Tubulin (Sigma, T6793), mouse anti-γ-Tubulin (Sigma, T5326), mouse anti-PCM1 (Santa Cruz, SC-398365), mouse anti-human DNAI2 (Abnova, H00064446-MO1), mouse anti-α-Tubulin (Sigma, T5168), mouse anti-HA (Santa Cruz, SC7392), mouse anti-GFP (Abcam, ab1218), rabbit anti-GFP (Santa Cruz SC-8334) mouse anti-MYC (Santa Cruz, SC-40), rabbit anti-HA (Santa Cruz, SC-805) and rabbit anti-MYC (Santa Cruz, SC-289). Antibodies to DNAH5 and DNAH9 were kindly provided by H. Takeda, described previously [41]. Rabbit antibodies to TTC25 were described previously [42]. Alexa Fluor-conjugated secondary antibodies for immunofluorescence staining were obtained from Invitrogen.

### Immunofluorescence staining

E8.0 embryos were collected in ice-cold phosphate buffered saline (PBS) and fixed in 4% paraformaldehyde for 30 min on ice. They were then dehydrated at -20˚C in methanol, exposed to a blocking reagent (Perkin Elmer FP1020), for 1 h at 4˚C, incubated overnight at 4˚C with primary antibodies diluted in blocking reagent (Perkin Elmer FP1020), washed for 6 h with PBS containing 0.1% Triton X-100 (PBST), incubated overnight at 4˚C with Alexa Fluor–conjugated secondary antibodies diluted in blocking reagent (Perkin Elmer FP1020), solution or Can Get Solution TM A/B (Toyobo), and washed again. The nodes of the stained embryos were excised and placed on glass slides fitted with silicone rubber spacers, and covered with cover glass.

Tracheae of adult mice were isolated, cut longitudinally in ice-cold PBS, fixed in 4% paraformaldehyde for 1 h at 4°C, permeabilized with 1% PBST for 10 min at 4°C, and then processed for immunostaining as for the node. The tissue was then placed on a glass slide with silicone rubber spacers and covered with a cover glass. For preparation of cryosections of the immunostained trachea, samples were exposed overnight at 4°C to sucrose at increasing concentrations of up to 30%, embedded in OCT compound, frozen in liquid nitrogen, and sectioned at a thickness of 10 μm. The sections were dried, washed with PBS, mounted in Antifade Prolong Gold (Invitrogen), and covered with a cover glass.

Tracheal epithelial cells, isolated as described previously [43], were fixed in methanol for 10 min at –20°C, exposed to 1% bovine serum albumin at room temperature for 30 min, and incubated for 1 h at room temperature with primary antibodies diluted in Can Get Solution TM A (Toyobo). After washing with PBST, the cells were incubated for 1 h at room temperature with Alexa Fluor-conjugated secondary antibodies diluted in Can Get Solution A and then washed again. All immunostained samples were observed with Olympus FV1000, FV3000 or Zeiss LSM 880 with Airyscan microscopic systems.

## Imaging of ciliary motion

Trachea and brain were collected into HEPES-buffered Dulbecco's modified Eagle's medium (DMEM) or high-glucose DMEM (Nacalai), respectively. Embryos were collected into DMEM-HEPES with 10% Fetal bovine serum (FBS). Motility of cilia was examined at 25°C with a high-speed CMOS camera (HAS-500M or HAS-U2, DITECT) at a frame rate of 100 frames/s for node cilia or 500 frames/s for tracheal and brain ependymal cilia. The pattern of ciliary motion was traced and analyzed with ImageJ and Photoshop CC (Adobe).

## TEM

E8.0 embryos and adult trachea were fixed with 2% paraformaldehyde and 2.5% glutaraldehyde (and 0.1% tannic acid for the trachea) in 0.1 M cacodylate buffer. The samples were then washed with 4% sucrose in 0.1 M cacodylate buffer, exposed to 1% osmium tetroxide, stained en bloc with 0.5% uranyl acetate, dehydrated with a graded series of ethanol, and embedded with the use of a Poly/Bed 812 Embedding Kit (Polysciences). Ultrathin (60 nm) sections were prepared with a diamond knife, mounted on 200-mesh copper grids, stained with 2% uranyl acetate and 3% lead citrate, and observed with a JEM-1400 plus microscope (JEOL).

## Immunoblot analysis

Ciliated cell cortices were prepared from the mouse trachea as described previously [44], with some modifications. In brief, cilia were removed by vortex mixing in a solution containing 20 mM Hepes-NaOH (pH 7.5), 10 mM $CaCl_2$, 1 mM EDTA, 50 mM NaCl, 1 mM dithiothreitol and cOmplete protease inhibitor cocktail (Roche), and the isolated cilia were resuspended in a solution containing 20 mM Hepes-NaOH (pH 7.6), 50 mM KCl, 4 mM $MgSO_4$, 0.5 mM EDTA, 1 mM dithiothreitol, and cOmplete (Roche). The membrane of the cilia was removed by exposure to 0.1% Triton X-100 for 10 min on ice, and the axonemes were then washed with resuspension buffer and treated for 10 min on ice with resuspension buffer supplemented with high salt (0.6 M NaCl or NaI). The high salt-treated axonemes were subjected to centrifugation at 15,000 x *g* for 12 min at 4°C, and the resulting pellet and supernatant were subjected to immunoblot analysis with antibodies to CFAP53, to DNAIC2, and to α-Tubulin. Immune complexes were detected with horseradish peroxidase-conjugated antibodies to rabbit (NA934, GE HealthCare) or mouse (NA931, GE HealthCare) immunoglobulin G.

## Co-immunoprecipitation analysis

The coding sequence of mouse *Cfap53* was cloned into the pxj40-HA vector (Roche), whereas those of mouse *Dnaic1*, *Dnaic2*, *Dynll1*, *Ccdc114*, *Ccdc151* and *Ttc25* were cloned into pxj40-MYC (Roche) and *Dnah11*(N-terminus 1-1000aa), *Dnah11*(N-terminus 1001-1700aa) were cloned into pxj40-GFP (Roche). The indicated combinations of plasmids were introduced into HEK293T cells in 10-cm dishes by transfection for 24 h with the use of the Lipofectamine 2000 reagent (Thermo Fisher Scientific). The cells were then lysed in 1 ml of RIPA buffer (Thermo Fisher Scientific) supplemented with cOmplete Mini protease inhibitors (EDTA-free, Roche) and subjected to immunoprecipitation with 2 µg of mouse monoclonal antibodies to HA, MYC or GFP together with 30 µl of protein A-agarose beads (Roche). The resulting immunoprecipitates as well as the cell lysates were then subjected to immunoblot analysis with rabbit polyclonal antibodies to HA, MYC or GFP. Immune complexes were detected with horseradish peroxidase-conjugated secondary antibodies to mouse (Promega, W4028) or rabbit (Promega, W4018) immunoglobulin G.

## Supporting information

**S1 Fig. The length of node cilia and tracheal cilia from the *Cfap53⁻/⁻* mouse.** (A) The length of node cilia from *Cfap53⁺/⁺* and *Cfap53⁻/⁻* embryos. Data are presented as mean ± SD (n = 20 independent variables); two tailed Student's t-test (*p = 0.0109). (B) The length of tracheal cilia from *Cfap53⁺/⁺* and *Cfap53⁻/⁻* mice. Data are presented as mean ± SD (n = 16 independent variables); two tailed Student's t-test (n.s.; p = 0.6608). Note that mutant node cilia are slightly shorter but mutant tracheal cilia retained the normal length.
(TIF)

**S2 Fig. TEM images of sperm from *Cfap53⁺/⁺* and *Cfap53⁻/⁻* male mice.** Sperm from *Cfap53⁺/⁺* and *Cfap53⁻/⁻* male mice was subjected to TEM analysis. Note that normal axonemal structure is severely disrupted in the mutant sperm. Arrows indicate the center of the cross section of sperm. Note that the 9+2 structure is observed in *Cfap53⁺/⁺*, but is disrupted in *Cfap53⁻/⁻*. N: Nucleus, DF: Outer dense fiber, MI: Mitochondrion. Scale bars, 500 nm.
(TIF)

**S3 Fig. The CFAP53::Venus fusion protein is functional.** (A) Construction of a transgenic mouse strain harboring a *Cfap53^Venus* BAC. The coding sequence for Venus together with that for a (Gly₄Ser)₃ linker and poly(A) (pA) sequence as well as a loxP site was inserted into exon 8 of *Cfap53* gene immediately upstream of the TAG stop codon. ORF, open reading frame; UTR, untranslated region. (B) Copy number of the BAC transgene was examined for two transgenic lines (#19, #66) harboring the *Cfap53^Venus* BAC by quantitative PCR. Genomic DNA from *Dnah11^Venus/Venus* mouse, which has two copies of *Venus*, served as a control. Note that lines #19 and #66 have 2 copies and 1 copy of the transgene, respectively. (C) Genotyping of *Cfap53⁻/⁻* with *Cfap53^Venus*. *Cfap53⁻/⁻*, exon 2 excluded, have two alleles of different length. The male, *Cfap53⁺/⁻* (long alleles: used in other experiments) with *Cfap53^Venus*, was crossed with females, *Cfap53⁺/⁻* (short alleles). We can distinguish between *Cfap53⁻/⁻*, *Cfap53^Venus* and *Cfap53⁺/⁻*, *Cfap53^Venus* by length of *Cfap53⁻/⁻* alleles in F1 mice. (D) Mice genotyped in (C) at 4 weeks of age. Hydrocephalus apparent in *Cfap53⁻/⁻*(KO) mice was rescued by introduction of the *Cfap53^Venus* transgene.
(TIF)

**S4 Fig. Localization of CFAP53::Venus fusion protein in tracheal and node cilia.** (A–D) Immunofluorescence staining of the node of *Cfap53^Venus* (A and C) or control (B and D)

embryos at E8.0 with antibodies to GFP (A and B) and to acetylated (acet.) Tubulin (C and D). Scale bar, 10 μm. (E–G) Higher magnification images of a node cilium of a *Cfap53*$^{Venus}$ embryo showing GFP, acetylated Tubulin, and merged staining, respectively. CFAP53::Venus was predominantly localized at the base of node cilia but was also detected at a much lower level in the axoneme. Scale bar, 1 μm. (H–K) Immunofluorescence staining of the trachea of adult *Cfap53-*$^{Venus}$ and control mice with antibodies to GFP (H and I) and to acetylated (acet.) Tubulin (J and K). Scale bars, 20 μm. Insets in H, I show higher magnification views of tracheal cilia. Scale bars, 5 μm. (L–Q) Immunofluorescence analysis of an isolated tracheal MCC from a *Cfap53*$^{-/-}$ mouse harboring the *Cfap53*$^{Venus}$ transgene (O-Q) or a *Cfap53*$^{+/+}$ mouse without the transgene (L-N). CFAP53::Venus was absent in the distal region of tracheal cilia (O-Q). Scale bars, 5 μm. (R-T) Immunofluorescence staining of the node cilia of *Cfap53*$^{Venus}$ embryos at E8.0 with antibodies to GFP (R) and to PCM1 (S). Scale bar, 1 μm. (U-W) Immunofluorescence staining of the node cilia of *Cfap53*$^{Venus}$ embryos at E8.0 with antibodies to GFP (U) and to γ-Tubulin (V). Scale bars, 1 μm.
(TIF)

**S5 Fig. Construction of a *Dnah11*$^{Venus}$ allele encoding a functional DNAH11::Venus fusion protein.** (A) Construction of a transgenic mouse strain harboring a *Dnah11*$^{Venus}$ allele. The targeting vector contained a *neo* cassette and loxP sites, and the coding sequence for Venus was inserted into exon 43 (which encodes the second AAA motor domain) of *Dnah11* together with that for (Gly$_4$Ser)$_3$ linkers. (B) Laterality of milk spots for WT, *Dnah11*$^{Venus/Venus}$, and *Dnah11*$^{iv/iv}$ (homozygous for a spontaneous *Dnah11* mutation that results in *situs inversus*). All milk spots of *Dnah11*$^{Venus/Venus}$ mice were located on the left (normal) side (n = 39/39), whereas those of *Dnah11*$^{iv/iv}$ mice were L-R randomized (n = 3/8). (C) Live fluorescence imaging of DNAH11::Venus (green) in adult tracheal cilia of mice harboring a *Dnah11*$^{Venus}$. DNAH11::Venus was detected in proximal region of tracheal cilia. BF, bright-field. D and P in the merged image indicate the distal and proximal region of cilia, respectively. Scale bar, 5 μm.
(TIF)

**S6 Fig. Localization of DNAH9 in ciliated cells of the trachea and node.** (A-F) Immunofluorescence staining with antibodies to DNAH9 (green) and to acetylated (acet.) Tubulin (magenta) of the node at E8.0 and adult trachea (G-L) of *Cfap53*$^{-/-}$ and control mice. DNAH9 was not detected in node cilia of *Cfap53*$^{+/-}$or *Cfap53*$^{-/-}$embryos (A-F), whereas its expression was maintained in tracheal cilia of the mutant (G-L). Scale bars, 10 μm (A-F) or 2 μm (G-L).
(TIF)

**S7 Fig. DNAH9 is present in tracheal cilia but not in node cilia.** (A) *Dnah9*$^{Venus}$ allele was generated by inserting (G$_4$S)$_3$ Venus at the carboxyl terminus of DNAH9. (B) Live fluorescence imaging of DNAH9::Venus (green) in the node at E8.0 and in adult tracheal cilia of mice harboring a *Dnah9*$^{Venus}$ transgene. DNAH9::Venus was detected in tracheal cilia but not in node cilia. The dashed lines indicate the outline of the node. Bright-field (BF) and merged images are also shown. Scale bars, 10 μm. (C) Immunofluorescence staining with antibodies to GFP (green) and to acetylated (acet.) Tubulin (magenta) of adult trachea from *Dnah9*$^{Venus}$ mice. Note that DNAH9::Venus is preferentially localized to the distal region of tracheal cilia. D and P indicate and distal and proximal regions of tracheal cilia, respectively. Quantitative analysis confirms a higher intensity of GFP signals in the distal region. Data are presented as mean ± SD (n = 3 independent variables); two tailed Student's t-test (*p = 0.0144). Scale bar, 5 μm.
(TIF)

**S8 Fig. TEM images of *Cfap53*<sup>-/-</sup> tracheal cilia at different proximal-distal levels.** (A) Serial TEM sections were made from *Cfap53*<sup>-/-</sup> tracheal cilia along the proximal-distal axis. The level of each section shown in (B-N) is illustrated. (B-H) Lower magnification images of seven sections. Scale bar, 1 μm. (I-N) Higher magnification images. Scale bar, 100 nm. The cilium from which the high magnification image is derived is indicated at the right-top of each panel. For example, panel (I) shows a section image of cilium 1 at the level indicated in (A). Arrowheads denote microtubules lacking ODA.
(TIF)

**S9 Fig. Association of CFAP53 with DNAIC proteins and TTC25.** (A-B) HEK293T cells transfected with expression vectors for HA-tagged CFAP53 or MYC epitope–tagged DNAIC1, DNAIC2, DYNLL1 (cytoplasmic dynein light chain), TTC25 (or with the corresponding empty vectors), as indicated, were subjected to immunoprecipitation (IP) with antibodies to HA or to MYC, and the resulting precipitates as well as the original whole cell lysates (WCLs) were subjected to immunoblot analysis with antibodies to HA or MYC. CFAP53 interacted specifically with the axonemal dyneins DNAIC1, DNAIC2 and the DC member TTC25.
(TIF)

**S10 Fig. TTC25 localizes to the axoneme of tracheal cilia in *Cfap53*<sup>−/−</sup> mice.** Tracheal cilia of adult *Cfap53*<sup>−/−</sup> and control mice were subjected to immunofluorescence staining with antibodies to TTC25 (green) and to acetylated (acet.) Tubulin (magenta). TTC25 was maintained in tracheal cilia of *Cfap53*<sup>−/−</sup> mice, albeit at a reduced level. Scale bars, 5 μm.
(TIF)

**S11 Fig. Search for ciliary protein genes expressed in tracheal ciliated cells but not in node ciliated cells.** Total RNA isolated from adult trachea or E8.0 node of mice was subjected to RT-PCR analysis of the 18 indicated genes. Amplified cDNAs were detected by agarose gel electrophoresis. Representative results for *Efhc1* and *Cfap20* are shown. No gene was found to be specifically expressed in tracheal ciliated cells and not in node ciliated cells.
(TIF)

**S1 Video. Motion of node cilia in *Cfap53*<sup>+/-</sup> embryo shown in Fig 2F.** Note that node cilia are motile. The images were captured at 100 frames/s for 2 s. Scale bar is 10 μm.
(MP4)

**S2 Video. Motion of node cilia in *Cfap53*<sup>-/-</sup> embryo, shown in Fig 2F.** Note that node cilia are immotile. The images were captured at 100 frames/s for 2 s. Scale bar is 10 μm.
(MP4)

**S3 Video. Motion of tracheal cilia of *Cfap53*<sup>+/+</sup> and *Cfap53*<sup>-/-</sup> mice shown in Fig 2G.** Note that the shape of a cilium is superimposed on the last beat. The images were captured at 200 frames/s for 1 s.
(MP4)

**S4 Video. Motion of brain ventricle cilia of *Cfap53*<sup>+/+</sup> and *Cfap53*<sup>-/-</sup> mice shown in Fig 2G.** Note that the shape of a cilium is superimposed on the last beat. The images were captured at 200 frames/s for 0.75 s.
(MP4)

**S5 Video. Motility of sperm from *Cfap53*<sup>+/+</sup> mice.** Sperm can swim normally. The images were recorded at 200 frames per second (fps) and playing video speed is 20 fps.
(MPEG)

**S6 Video. Motility of sperm from *Cfap53*<sup>-/-</sup> mice.** Sperm are immotile. The images were recorded at 200 frames per second (fps) and playing video speed is 20 fps.
(MPEG)

**S7 Video. Motility of tracheal cilia from *Dnah9*<sup>+/+</sup> mice shown in Fig 6F.** Motility of tracheal cilia was recorded at 200 frames per second (fps) and playing video speed is 20 fps.
(AVI)

**S8 Video. Motility of tracheal cilia from *Dnah9*<sup>-/-</sup> mice shown in Fig 6F.** Motility of tracheal cilia was recorded at 200 frames per second (fps) and playing video speed is 20 fps.
(AVI)

**S9 Video. Motility of brain ventricle cilia from *Dnah9*<sup>+/+</sup> mice shown in Fig 6F.** Motility of brain ventricle ependymal cilia was recorded at 200 frames per second (fps) and playing video speed is 20 fps.
(AVI)

**S10 Video. Motility of brain ventricle cilia from *Dnah9*<sup>-/-</sup> mice shown in Fig 6F.** Motility of brain ventricle ependymal cilia was recorded at 200 frames per second (fps) and playing video speed is 20 fps.
(AVI)

**S11 Video. Motility of sperm from *Dnah9*<sup>+/+</sup> mice.** Motility of sperm was recorded at 200 frames per second (fps) and playing video speed is 20 fps.
(AVI)

**S12 Video. Motility of sperm from *Dnah9*<sup>-/-</sup> mice.** Motility of sperm was recorded at 200 frames per second (fps) and playing video speed is 20 fps.
(AVI)

## Acknowledgments

We thank H. Takeda (University of Tokyo) for anti-DNAH5 and anti-DNAH9 antibodies, M. Brueckner for DNAH11 cDNA clones, and Sachiko Tsukita (Osaka University) for invaluable advice on immunostaining.

## Author Contributions

**Conceptualization:** Takahiro Ide, Sudipto Roy, Hiroshi Hamada.

**Data curation:** Takahiro Ide, Wang Kyaw Twan, Hao Lu, Yayoi Ikawa, Nicole Henninger.

**Formal analysis:** Takahiro Ide, Wang Kyaw Twan, Hao Lu.

**Funding acquisition:** Sudipto Roy, Hiroshi Hamada.

**Investigation:** Takahiro Ide, Wang Kyaw Twan, Hao Lu, Yayoi Ikawa, Lin-Xenia Lim, Nicole Henninger, Xiumin Yan, Hidetaka Shiratori.

**Methodology:** Takahiro Ide, Wang Kyaw Twan, Hao Lu, Katsuyoshi Takaoka, Vijay Narasimhan.

**Project administration:** Sudipto Roy, Hiroshi Hamada.

**Resources:** Yayoi Ikawa, Hiromi Nishimura, Katsuyoshi Takaoka, Vijay Narasimhan, Xiumin Yan, Hidetaka Shiratori, Hiroshi Hamada.

**Software:** Takahiro Ide, Wang Kyaw Twan.

**Supervision:** Sudipto Roy, Hiroshi Hamada.

**Validation:** Sudipto Roy, Hiroshi Hamada.

**Visualization:** Takahiro Ide, Wang Kyaw Twan.

**Writing – original draft:** Takahiro Ide, Wang Kyaw Twan, Hiroshi Hamada.

**Writing – review & editing:** Sudipto Roy, Hiroshi Hamada.

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
