## [Decision Letter · Decision Letter 0]

25 May 2020

Dear Dr Hamada,

Thank you very much for submitting your Research Article entitled 'Differential requirement of Cfap53 for ciliary motility in 9+0 and 9+2 motile cilia' to PLOS Genetics. Your manuscript was fully evaluated at the editorial level and by independent peer reviewers. The reviewers appreciated the attention to an important problem, but raised some substantial concerns about the current manuscript. Based on the reviews, the editors would be willing to consider a new manuscript in which the current work serves as a starting point, and that includes adressing several points raised by the reviewers.  These include addressing 1.) competition between the tagged and untagged versions of proteins, 2.) addressing localization of the ODA by TEM with respect to the doublets and the proximal/distal position, 3.)  one reviewer felt that data were lacking for a role of cFAP53 in IFT transport , and 4.) the proximal localization of cFAP53.  It is important to address all of the points raised by the reviewers.  We cannot, of course, promise publication at that time. 

If you decide to revise the manuscript for further consideration at PLOS Genetics, please aim to resubmit within the next 60 days, unless it will take extra time to address the concerns of the reviewers, in which case we would appreciate an expected resubmission date by email to plosgenetics@plos.org.

[LINK]

We are sorry that we cannot be more positive about your manuscript at this stage. Please do not hesitate to contact us if you have any concerns or questions.

Yours sincerely,

Susan K. Dutcher

Associate Editor

PLOS Genetics

Gregory Barsh

Editor-in-Chief

PLOS Genetics

Reviewer's Responses to Questions

**Comments to the Authors:**

Reviewer #1: see attached file

Reviewer #2: Summary:

Ide and colleagues characterize Cfap53-/- mice in conjunction with other ciliary mouse models (e.g., Dnah9-/-, Rsph4a-/-, etc.) and propose a mechanism for how Cfap53 can differentially regulate nodal / tracheal / ependymal ciliary beating. A particular strength of the study is the characterization of abnormal ciliary waveforms in these ciliated cell types and quantitative analysis of how outer dynein arms are differentially affected in nodal and tracheal cilia. Overall, this study provides novel and interesting insights that advance understanding of the CFAP53 gene and suggest distinct mechanisms that underlie 9+2 as opposed to 9+0 ciliary beating. This manuscript is suitable for PLoS Genetics as the authors use a series of novel, genetically stable ciliary mutants to address how ciliary beating differs among these cell types. However, several concerns need to be adequately addressed and the manuscript extensively revised before it is suitable for publication.

Major concerns:

CFAP53 (also known as CCDC11) has been reported to cause body laterality defects without the typical respiratory symptoms consistent with a diagnosis for primary ciliary dyskinesia (Narasimhan et al., 2015). In the aforementioned study, anti-human CFAP53 antibody was used to demonstrate localization throughout the entire axoneme of respiratory cilia – the specificity was established by its absence in respiratory cilia from a patient with a homozygous loss-of-function CFAP53 mutation. Here, Ide and co-authors describe mouse Cfap53 localization in pericentriolar satellites in nodal cilia and proximal localization in tracheal cilia. The most convincing data is the localization of Cfap53-Venus at the pericentriolar satellites of nodal cilia, where there is a clear colocalization (yellow overlay) with PCM-1 and gamma tubulin (Fig 3). This Cfap53 localization around the pericentriolar region is appreciated in three separate images (Fig. 3E, 3O, and 3R). However, the localization of CFAP53::Venus in nodal cilia also appears as discrete puncta in the distal ciliary region (see Fig. 3O-Q and Fig. 3R-T) as well as accumulation around the ciliary base. This has significant implications for interpreting the mechanism whereby Cfap53 differentially regulates ODAs by either facilitating its transport into nodal cilia or selectively stabilizing ODAs (position 3 and 8) in the axonemes of tracheal cilia.

The authors also note that Cfap53::Venus localized to the proximal ciliary region in respiratory axonemes (Fig. 3). This interpretation is less convincing and appears to conflict with human CFAP53 data, which shows its native detection along the entire respiratory axoneme (Narasimhan 2015). The localization of Cfap53::Venus using anti-GFP antibody in trachea appears diffuse with more intense signal in the proximal region but also signal in the distal ciliary region. This is in contrast to DNAH11, which shows a clearly defined reactivity in only the proximal ciliary region. The conclusion that Cfap53 is axonemal in tracheal cilia (and possibly nodal cilia) seems sufficient and is consistent with other reports describing CFAP53 in other species, whereas the interpretation that Cfap53 has a proximal localization in mouse trachea requires further clarification as to how / why this would be different. This latter point would be significantly strengthened if the authors can specify whether, in the course of quantifying ODAs in TEMs of Cfap53-mutant cilia, the deficiency of ODAs at positions 3 and 8 (or elsewhere) were in the proximal ciliary region of trachea axonemes as opposed to the distal ciliary region. Furthermore, there is no presentation of native Cfap53 localization by immunofluorescence microscopy, which would be very helpful to corroborate the localization pattern of Cfap53-Venus using anti-GFP antibody. Can the authors comment on why anti-Cfap53 protein (which is included in the material and methods section) is not shown / tested for immunofluorescence microscopy? Have the authors used the published anti-human CFAP53 antibody in Narasimhan 2015 to test cross-reactivity with mouse axonemes?

Given the points raised above and the interesting result that a subset of ODAs are preferentially affected in tracheal but not nodal cilia, it is conceivable that Cfap53 axonemal localization is restricted to a subset of microtubule doublets (positions 3 and 8) in trachea cilia but targeted to most or all microtubule doublets in nodal cilia. In this alternate model, Cfap53 is axonemal in both tracheal and nodal cilia and predicts that subsets of MTDs are qualitatively different between these cell types. This is not without precedent, as MTDs at position 5 and 6 have a unique connecting structure that is not present among the other MTDs.

Minor Concerns:

References

- Page 11, lines 209-211: the study by McGrath et al., 2003 (reference 29) does not distinguish / define proximal ciliary region in their study whereas the DNAH11 study from the group of Omran (reference 11) is the first to make this distinction. However, both of these studies note localization of Gfp-Dnah11 throughout the entire nodal cilia axoneme. This sentence should be rephrased to more accurately reflect these points.

Phrasing should be modified

- Page 5, lines 63 – 66: “… those with situs inversus and heterotaxy are typically afflicted exclusively with aberrations in laterality…” This suggests that most cases of situs inversus and heterotaxy occur in isolation but this is not entirely accurate as most cases of laterality cases are associated in conjunction with PCD. This should be reworded to convey that there are certain cases / reports of situs inversus / heterotaxy with no accompanying diagnosis of PCD.

The description of Dnah9-/- mice is novel and regarding analysis of laterality, the authors state that 2/2 embryos showed normal situs composition. This is a low number to strictly rule out randomization of laterality and especially heterotaxy – if the authors have not analyzed additional embryos it would be helpful to state that laterality defects especially heterotaxy cannot be fully excluded with the analysis of only 2 embryos. This would be compatible with Dnah9 mRNA protein expression at the node and the possibility that loss of Dnah9 in mice could have very subtle / low frequency effect on laterality development including heterotaxy. Have the authors tested anti-Dnah9 antibody by Western blotting to support its specificity?

Regarding the speculation that outer dynein arms in nodal cilia may be composed of a hetero-trimeric complex (page 18, lines 380-383), is there any published evidence to support the existence of three-headed dynein arm complexes in mammalian cilia?

The title does not appreciably convey the novelty of the findings, as previous publications have already demonstrated a differential role for CFAP53 in tracheal versus nodal ciliary motility. This could be strengthened by possibly mentioning selective versus complete ODA deficiency between these cell types.

As the authors also indicate Cfap53 expression in male testes, can they comment on the fertility status of Cfap53-/- males?

In Figure 9, the authors state that Ttc25 is lost in Cfap53-/- node cilia. However, considering the inset highlighting the cilia, the signal looks much weaker for Ttc25 in heterozygous control (Fig. 9J) than in homozygous null (inset Fig.9M), and cilia are more easily discerned with even some level of colocalization in this sample (Fig. 9O). Can the authors comment further on this?

Regarding the model of Cfap53 in motile cilia (Fig. 10), point (B) refers to interaction of Cfap53 with IFT components (page 34). Is there published evidence for the interaction of Cfap53 with IFT components?

The authors conclude that Cfap53 interacts with the ODA docking machinery, specifically TTC25, as well as dyneins associated with the outer dynein arms (Dnai1, Dnai2, Dnah11). There are no control lanes for Western blots in Fig. 8C and 8G, which ideally should be compared on the same blot / exposure to interpret strength of signal relative to control. The interaction of Dnah11 fragment aa 1000-1700 with Cfap53 is extremely weak (Fig. 8D) and should be interpreted with caution, as this fragment was more robustly immunoprecipitated than the aa 1-1000 construct (compare band intensities of anti-GFP with Dnah11 1-1000 and Dnah11 1000-1700, top panel) but shows a much weaker / almost negligible signal for Cfap53 (middle panel, anti-HA antibody). Additionally, the expected molecular weight of Ttc25 is approximately 72 kDa, however the blots (Fig. 8E, top and bottom panels) indicate a migration closer to 50 kDa – can the authors provide further information including whether a smaller isoform of Ttc25 was tested?

Regarding the use of NaI solution to discriminate proteins tightly associated with axonemal microtubules, are there published references using this technique to analyze axonemes in other ciliary models / species?

The combined phenotype of Cfap53-/- + Rsph4a-/- mice show a reduction in ODAs in tracheal cilia (Fig. 7M). This is an interesting result but also unexpected as radial spoke proteins are not expected to interact with outer dynein arm proteins. Can the authors further comment on how this synergy may regulate ODAs and also if there are previous studies that support a functional association between radial spoke and outer dynein arms?

Cfap53 has been proposed to function as a microtubule-interacting protein within the A-tubule of Chlamydomonas axonemes. If Cfap53 is strictly confined to the inside of the A-tubule, how are interactions with dynein associated components (e.g., Ttc25, Dnah11) possible? Is Cfap53 in some way accessible to the surface of the A-tubule? Also, MIPs have been proposed to confer stability to the 9+0 axoneme, yet the significant phenotype in Cfap53-/- axonemes is partial or complete ODA deficiency (trachea and node, respectively) with no indication of shorter / less stable axonemal MTDs. Can the authors comment on morphology / stability of Cfap53-/- node cilia compared to control? Are their differences between Chlamydomonas and mouse axonemes that can help explain these differences?

Reviewer #3: OVERVIEW:

The molecular and structural differences underlying mammalian motile cilia diversity in terms of function and waveform is largely unknown. Here, Ide, Twan and Lu et al have carried out a detailed functional and molecular analysis of the coiled-coil domain containing protein CFAP53 in mammalian motile cilia. Homozygous crispant null alleles of CFAP53 disrupt ciliary beat waveform in the MCCs of the ependymal and respiratory epithelia and show reduced beat frequency in tracheal cells. In contrast, structure and motility of the nodal (9 + 0) cilia is more strongly affected- by TEM that ODAs are heavily depleted. This likely underlies the isolated situs defects, as opposed to more syndromic primary ciliary dyskinesia (PCD), previously published for human patients. Even in the mildly affected MCC axonemes from Cfap53-/- mice, the authors could demonstrate ODAs on specific doublets were preferentially lost, those in the plane of beating in 9 + 2 axonemes. This hypothesis is elegantly test by generating Rsph4a-/- Cfap53+/- - which now show complete loss of ODAs across the field in these rotational mutant MCCs.

The authors have developed an impressive arsenal of BAC transgenic mouse lines to monitor expression and localization of gene of interest CFAP53 as well as two ODA dynein heavy chain tagged lines (DNAH9 and 11). Beyond this study, these reagents are very useful tools for those working in the mammalian motile cilia field. In their BAC transgenics, they show the expression of Cfap53 transcripts in mouse motile ciliated tissues of the mouse and importantly demonstrate that their CFAP53-Venus fusion is functional- rescuing the hydrocephaly phenotype of Cfap53 mutant mice. Using endogenous tagging approaches for axonemal dynein heavy chain fusion lines- they show that nodal but not tracheal cells lack ODA heavy chain staining. Moreover, they suggest that that unlike humans, mice seem to lack DNAH9 from the nodal cilia and do not require it for correct organ positioning (n=2 embryos).

Finally, in fractionation studies, they demonstrate CFAP53 is associated with microtubule axoneme under conditions where ODAs become detached. In a series of transient transfections in HEKs to pulldown interactors of CFAP53- they demonstrate that CFAP53 interacts with selected ODA proteins and the docking complex protein TTC25 but not more distal CCDC114 or 151. They further demonstrate that axonemal localisation of TTC25 is reliant on CFAP53, but not vice versa. This leads to their model which suggests that CFAP53 functions to help transport ODAs into the axoneme and assists in their docking there and that this function is redundant in respiratory cilia where is likely another protein compensates for CFAP53.

Overall, it is a very elegant mouse genetics paper to functionally dissect roles for known human disease gene CFAP53 in diverse mammalian cilia types. The motile-specific phenotypes are unexpected given the KD/morphant phenotypes from other species and mammalian cell culture, where a role in ciliogenesis was reported. However, this story is much cleaner in terms of explaining the phenotypes in patients with CFAP53 variants. Overall, the work is of very high quality, careful analysis and broad significance to the cilia community. Major points are for proper analysis of data presented (quantitation, expanded analysis and/or discussion).

Major points:

1) BAC generation and experiments: While BAC clones are useful for preserving endogenous regulatory elements- levels and patterns of expression- I think these CFAP53 localization experiments were done in the background of wild type CFAP53. Given this, I would caution a little- dampen the emphasis made on differential localizations of CFAP53 between different motile cilia types. ‘Competition’ between tagged and untagged isoforms for cilia localization could be limited- we know this is the case for IFT proteins. You get higher labelling density within cilia if the tagged-isoform is the only one expressed. One workaround would be to validate using the endogenous CFAP53 expression using the antibody they have generated- for Fig8 if it works by IF.

i. Regardless, with the current images, quantitative image analysis of the distributions would be very helpful- average arbitrary units of fluorescence down the axonemes of different cilia types.

ii. Also, I did not find the numbers of BAC integrations (often concatomers, as well as locus effects) expression relative to endogenous locus. If the authors have this information- authors should include in the methods how many founders, how many copies, relative expression, screening?

iii. The literature suggests that CFAP53 is widely expressed in primary ciliated cells where it seems to function in ciliogenesis (i.e. Silva et al 2015)- however none of the previous work in human cell lines RPE1 or Xenopus over-expression shows cilia localization. This is an important finding of the current study and discrepancy should be discussed a little more clearly. This is important for the field.

2) CFAP53 allele characterization: Minor point for two CRIPSR alleles- how do the genomic changes affect protein (i.e. premature termination, in frame deletion). Please add in this information to Fig.2 A. Were the authors able to validate these predictions with their endogenous antibody they use in Figure 8A? Is any protein still present- truncated proteins are not null alleles. This is important given the potential discrepancies with the morphant/KD experiments in Xenopus and zebrafish- Mitchell and Mahjoub, as well as Khoka labs, as well as mammalian cells where ciliogenesis is affected. This should be discussed.

3) DNAH9-Venus and DNAH11-Venus localizations: Beautiful endogenous tagging approaches for axonemal dynein heavy chain fusion lines will be great resources for the community. If this paper is to be where they are referenced/introduced, I suggest however that more image analysis is needed in terms of the proximal-distal intensities- could a plot of average axoneme distributions for Venus-tagged molecules be shown next to the image panels in Figure 5 and 6. For example, DNAH11 is expected to be proximal in its localization but in fact in 5G and 5J it looks like it is excluded (non-overlapping) with the AcTub staining. A graph capturing the signals would be much easier to interpret. Similarly, Fig6, DNAH9-Venus looks much more broadly than restricted to the distal tip. Moreover, do these boundaries change at all in the mutants? In the text authors make the statement (lines 218)- ‘distal localization of Dnah9 was maintained, although its level was reduced compared with that apparent in control mice’. There is no quantitation here- quantitating the signals similar to distribution is required if these statements are to be made. For example, 6J-L- if you averaged the DNAH11/acTub signal intensity, the apparent decreased in DNAH11 would likely not hold. Again quantitation for the nodal cilia would also be helpful- magnifications of the DNAH9-Venus nodal cilia would also be necessary.

Minor points:

1) Phenotype: CFAP53 is highly expressed in testes- is there a sperm defect suggest it is a trafficking module for the paralogues DNAH17 and DNAH8? Or is it not required? Similarly, given the hydrocephaly phenotype why not characterize ependymal distribution (localization and levels) of the heavy chains in these cilia, or TEM analysis? By only looking at two cilia types, it is possible that the respiratory cilia are the odd ones out and not the nodal cilia (i.e. ependymal cilia are also affected more severely or sperm motility). How penetrant is the gross hydrocephaly phenotype- 100% have domed heads or subclinical between animals? Along the same vein, which background were the animals made on and maintained on to understand this phenotype.

2) Nomenclature: Throughout mouse proteins should be in all caps (i.e. CFAP53) whereas genes italics title case (i.e. Cfap53). This needs careful and extensive changing throughout the manuscript- TTC25, DNAH9 , DNAH11 and intermediate chains please use DNAI1 and DNAI2, which is the correct form of the protein.

3) Figure 2H- the spread in the control frequencies is unusual- hyperkinetic to hypokinetic almost overlapping mutants- but it is a ‘pool’ of control animals-wild type and het. We recommend identifying genotypes by symbols- are the hets intermediate?

4) Along with major point 3, the language is currently strong in suggesting that that unlike humans, mice seem to lack DNAH9 from the nodal cilia and do not require it for correct organ positioning (n=2 embryos). Given the small numbers of Dnah9 mutants examined, I would temper conclusions to ‘suggesting DNAH9 is not expressed in, nor required for mouse nodal cilia motility’. Given mRNA expression by WISH, can the authors rule out issues with the signal-noise ratio for low protein expression may confound this interpretation for DNAH9? The DNAH11-Venus levels (GFP IF) are also significantly lower than say DNAH5 IF- the enrichment in the cilia not very apparent (see Figure 5A- cytoplasm to axoneme). A GFP-IP of E8.5 embryos pooled- no tagged protein pulled down? Something else to bolster. If not use language, like ‘appears’ and ‘suggests’.

5) Inner dynein arms: While the authors have demonstrated that there is an effect on ODAs in Cfap53 mutants, there is no mention of the effect on IDAs. While this is not necessarily expected given the evidence from patient cells it is nevertheless an important consideration. What would the mechanism be? This could be mitigated with expanded discussion on previous observations about IDAs and why you think it is right not to investigate them in this study. Quantitate IDAs from mutant TEM

6) Clarify Figure 9 shows some images of the node, however the cilia are very hard to see with the staining used. The labelling of G-I is also a bit unclear. Label G-I to indicate that these are Cfap53+/+ mice. Make the magnified boxes larger and make the entire image brighter so the cilia are easier to see.

7) Line 33-34: edit to ‘albeit with an altered beat pattern’

8) Centriolar satellites is used as plural by the field, even in cells with a single cilium like the node. They are highly dynamic and tend to migrate, fuse and divide in short time scales. See line 34 and throughout.

9) Lines 46-50: This sentence is too long. Split into 9+0 and 9+2 motile cilia sentences, each with a full stop.

10) Line 149: typo ‘lacking exon 2’.

11) Line 171: typo ‘that the Cfap53.’

12) Line 267: for clarity add ‘TT25 is thought to be most proximal to the microtubule’ currently unclear proximal to what.

13) Lines 355-359: The sentence needs work ‘It may also not necessarily be the issue of absence of a particular protein from 9+0 and presence of it in 9+2 cilia that confers the observed differences on the effects of the loss of CFAP53 on ODAs and DC in the two types of cilia, rather like CFAP53, a protein present in both cilia-types but with differential localization and function in each kind of cilia could also fulfil this role.’ Wouldn’t a differential localization and function likely involve the presence/absence of another protein to execute these changes- the logic is fuzzy.

14) Line 381-383: Isn’t it much more likely that there is a heterogenous mix heterodimers than a heterotrimer. i.e. not spatially separated dimers in the human node.

15) Line 448: Specify what is the ‘blocking solution’.

16) Line 451: Change to ‘Can Get SolutionTM A/B’, and on line 466.

17) Line 637: Change to ‘Figure Legends’

18) Line 704: typo ‘ODAs are…’

19) Line 754: reference for the interaction with IFTs if shown, if presumed state explicitly.

20) Line 781: provide n= for how many animals were examined for laterality defects in the legend.

21) Accessibility: most of the IF data is two color/merge- consider choosing red-green colour-blind friendly combinations. https://imagej.nih.gov/ij/docs/guide/146-9.html and https://www.ascb.org/science-news/how-to-make-scientific-figures-accessible-to-readers-with-color-blindness/

**Have all data underlying the figures and results presented in the manuscript been provided?**

Reviewer #1: Yes

Reviewer #2: None

Reviewer #3: Yes

PLOS authors have the option to publish the peer review history of their article (what does this mean?). If published, this will include your full peer review and any attached files.

Reviewer #1: No

Reviewer #2: No

Reviewer #3: No

---

## [Decision Letter · Decision Letter 1]

29 Oct 2020

Dear Dr Hamada,

We are pleased to inform you that your manuscript entitled "Cfap53 regulates mammalian cilia-type motility patterns through differential localization and recruitment of axonemal dynein components" has been editorially accepted for publication in PLOS Genetics. Congratulations!  Please make the minor changes suggested by Reviewer 3 before submitting the final version.

Yours sincerely,

Susan K. Dutcher

Associate Editor

PLOS Genetics

Gregory Barsh

Editor-in-Chief

PLOS Genetics

Comments from the reviewers (if applicable):

Reviewer's Responses to Questions

**Comments to the Authors:**

Reviewer #1: The authors have made a serious effort to address the comments and criticisms of all three reviewers. The most significant improvement has been the clarification about the localization of the three DHCs (DHC5, DHC9, and DHC11) in nodal versus tracheal cilia in the presence and absence of CFAP53. It remains unclear why loss of CFAP53 disrupts the assembly of DHC5 and DHC11 in nodal cilia, with no apparent effect on tracheal cilia.

Reviewer #2: The authors have significantly improved the manuscript and adequately addressed all concerns. Overall, this manuscript makes an important contribution to the literature, especially in the field of cilia and ciliopathies, and is thus strongly recommended for publication.“

Reviewer #3: The revised manuscript by Ide, Twan and Lu et al. is an elegant tour-de-force work for mammalian cilia motility- the authors should be very proud. The authors have addressed the majority of my concerns, aside from nomenclature which has been very confusing for the field. As such it is important that all the mouse proteins in this manuscript are listed according to proper nomenclature, in an attempt to promote consistency, which aids data retrieval and improves communication. Protein symbols are in capital letters for both human and mouse. Even JAX which they cite as a rebuttal clearly states ‘1.5.2 Protein symbols Protein designations follow the same rules as gene symbols, with the following two distinctions: Protein symbols use all uppercase letters.

Protein symbols are not italicized.’ http://www.informatics.jax.org/mgihome/nomen/gene.shtml#ps These rules are elaborated here in these references (Sundberg, J P, and P N Schofield. “Mouse genetic nomenclature. Standardization of strain, gene, and protein symbols.” Veterinary pathology vol. 47,6 (2010): 1100-4. doi:10.1177/0300985810374837; International Protein Nomenclature Guidelines https://www.ncbi.nlm.nih.gov/genome/doc/internatprot_nomenguide/)

Minor points- many of the figures are still red-green combinations- this ~10% of your audience won’t be able to fully appreciate your beautiful images. It is a little more effort but really encourages accessibility and inclusiveness of science.

Edits: Page 12, Line 243: instead of the human proteins in sperm given your species differences in heavy chains between cilia types, a more relevant reference would be Mali et al 2018 eLIFE doi: 10.7554/eLife.34389 (Figure 5D) which clearly shows which heavy chains are present in mice sperm and which are missing.

Figure 8 D,E: typo for Cfap53-HA (currently Cafp53)

Figure 10B: left panel add ‘s’ ‘centriolar satellites’

Supp Figure 3B: Venus should be capitalized on all your animal names for consistency

Supp Figure S9B: : typo for Cfap53-HA (currently Cafp53)

**Have all data underlying the figures and results presented in the manuscript been provided?**

Reviewer #1: Yes

Reviewer #2: Yes

Reviewer #3: Yes

PLOS authors have the option to publish the peer review history of their article (what does this mean?). If published, this will include your full peer review and any attached files.

Reviewer #1: No

Reviewer #2: No

Reviewer #3: No

**Data Deposition**

http://datadryad.org/submit?journalID=pgenetics&manu=PGENETICS-D-20-00551R1

**Press Queries**

---

## [Editor Report · Acceptance letter]

4 Dec 2020

PGENETICS-D-20-00551R1 

CFAP53 regulates mammalian cilia-type motility patterns through differential localization and recruitment of axonemal dynein components 

Dear Dr Hamada, 

We are pleased to inform you that your manuscript entitled "CFAP53 regulates mammalian cilia-type motility patterns through differential localization and recruitment of axonemal dynein components" has been formally accepted for publication in PLOS Genetics! Your manuscript is now with our production department and you will be notified of the publication date in due course.

With kind regards,

Livia Horvath

PLOS Genetics

On behalf of:
